# Image content is more important than Bouma's Law for scene metamers

**Thomas SA Wallis[1,2†]\*, Christina M Funke[1,2†], Alexander S Ecker[1,2,3,4], Leon A Gatys[1‡], Felix A Wichmann[5], Matthias Bethge[3,4,6]**

[1]Werner Reichardt Center for Integrative Neuroscience, Eberhard Karls Universität Tübingen, Tübingen, Germany; [2]Bernstein Center for Computational Neuroscience, Berlin, Germany; [3]Center for Neuroscience and Artificial Intelligence, Baylor College of Medicine, Houston, United States; [4]Institute for Theoretical Physics, Eberhard Karls Universität Tübingen, Tübingen, Germany; [5]Neural Information Processing Group, Faculty of Science, Eberhard Karls Universität Tübingen, Tübingen, Germany; [6]Max Planck Institute for Biological Cybernetics, Tübingen, Germany

**Abstract** We subjectively perceive our visual field with high fidelity, yet peripheral distortions can go unnoticed and peripheral objects can be difficult to identify (crowding). Prior work showed that humans could not discriminate images synthesised to match the responses of a mid-level ventral visual stream model when information was averaged in receptive fields with a scaling of about half their retinal eccentricity. This result implicated ventral visual area V2, approximated 'Bouma's Law' of crowding, and has subsequently been interpreted as a link between crowding zones, receptive field scaling, and our perceptual experience. However, this experiment never assessed natural images. We find that humans can easily discriminate real and model-generated images at V2 scaling, requiring scales at least as small as V1 receptive fields to generate metamers. We speculate that explaining why scenes look as they do may require incorporating segmentation and global organisational constraints in addition to local pooling.
DOI: https://doi.org/10.7554/eLife.42512.001

**\*For correspondence:**
thomas.wallis@uni-tuebingen.de

[†]These authors contributed equally to this work

**Present address:** [‡]Apple Inc, Cupertino, United States

## Introduction

Vision science seeks to understand why things look as they do (*Koffka, 1935*). Typically, our entire visual field looks subjectively crisp and clear. Yet our perception of the scene falling onto the peripheral retina is actually limited by at least three distinct sources: the optics of the eye, retinal sampling, and the mechanism(s) giving rise to crowding, in which our ability to identify and discriminate objects in the periphery is limited by the presence of nearby items (*Bouma, 1970*; *Pelli and Tillman, 2008*). Many other phenomena also demonstrate striking 'failures' of visual perception, for example change blindness (*Rensink et al., 1997*; *O'Regan et al., 1999*) and inattentional blindness (*Mack and Rock, 1998*), though there is some discussion as to what extent these are distinct from crowding (*Rosenholtz, 2016*). Whatever the case, it is clear that we can be insensitive to significant changes in the world despite our rich subjective experience.

Visual crowding has been characterised as compulsory texture perception (*Parkes et al., 2001*; *Lettvin, 1976*) and compression (*Balas et al., 2009*; *Rosenholtz et al., 2012a*). This idea entails that we cannot perceive the precise structure of the visual world in the periphery. Rather, we are aware only of some set of summary statistics or ensemble properties of visual displays, such as the average size or orientation of a group of elements (*Ariely, 2001*; *Dakin and Watt, 1997*). One of the appeals of the summary statistic idea is that it can be directly motivated from the perspective of efficient coding as a form of compression. Image-computable texture summary statistics have been shown to be correlated with human performance in various tasks requiring the judgment of peripheral

**eLife digest** As you read this digest, your eyes move to follow the lines of text. But now try to hold your eyes in one position, while reading the text on either side and below: it soon becomes clear that peripheral vision is not as good as we tend to assume. It is not possible to read text far away from the center of your line of vision, but you can see 'something' out of the corner of your eye. You can see that there is text there, even if you cannot read it, and you can see where your screen or page ends. So how does the brain generate peripheral vision, and why does it differ from what you see when you look straight ahead?

One idea is that the visual system averages information over areas of the peripheral visual field. This gives rise to texture-like patterns, as opposed to images made up of fine details. Imagine looking at an expanse of foliage, gravel or fur, for example. Your eyes cannot make out the individual leaves, pebbles or hairs. Instead, you perceive an overall pattern in the form of a texture. Our peripheral vision may also consist of such textures, created when the brain averages information over areas of space.

Wallis, Funke et al. have now tested this idea using an existing computer model that averages visual input in this way. By giving the model a series of photographs to process, Wallis, Funke et al. obtained images that should in theory simulate peripheral vision. If the model mimics the mechanisms that generate peripheral vision, then healthy volunteers should be unable to distinguish the processed images from the original photographs. But in fact, the participants could easily discriminate the two sets of images. This suggests that the visual system does not solely use textures to represent information in the peripheral visual field. Wallis, Funke et al. propose that other factors, such as how the visual system separates and groups objects, may instead determine what we see in our peripheral vision.

This knowledge could ultimately benefit patients with eye diseases such as macular degeneration, a condition that causes loss of vision in the center of the visual field and forces patients to rely on their peripheral vision.

DOI: https://doi.org/10.7554/eLife.42512.002

information, such as crowding and visual search (*Rosenholtz et al., 2012a*; *Balas et al., 2009*; *Freeman and Simoncelli, 2011*; *Rosenholtz, 2016*; *Ehinger and Rosenholtz, 2016*). Recently, it has even been suggested that summary statistics underlie our rich phenomenal experience itself—in the absence of focussed attention, we perceive only a texture-like visual world (*Cohen et al., 2016*).

Across many tasks, summary statistic representations seem to capture aspects of peripheral vision when the scaling of their pooling regions corresponds to 'Bouma's Law' (*Rosenholtz et al., 2012a*; *Balas et al., 2009*; *Freeman and Simoncelli, 2011*; *Wallis and Bex, 2012*; *Ehinger and Rosenholtz, 2016*). Bouma's Law states that objects will crowd (correspondingly, statistics will be pooled) over spatial regions corresponding to approximately half the retinal eccentricity (*Bouma, 1970*; *Pelli and Tillman, 2008*; though see *Rosen et al., 2014*). While the precise value of Bouma's law can vary substantially even over different visual quadrants within an individual (see e.g. *Petrov and Meleshkevich, 2011*), we refer here to the broader notion that summary statistics are pooled over an area that increases linearly with eccentricity, rather than the exact factor of this increase (the exact factor becomes important in the paragraph below). If the visual system does indeed represent the periphery using summary statistics, then Bouma's scaling implies that as retinal eccentricity increases, increasingly large regions of space are texturised by the visual system. If a model captured these statistics and their pooling, and the model was amenable to being run in a generative mode, then images could be created that are indistinguishable from the original despite being physically different (metamers). These images would be equivalent to the model and to the human visual system (*Freeman and Simoncelli, 2011*; *Wallis et al., 2016*; *Portilla and Simoncelli, 2000*; *Koenderink et al., 2017*).

*Freeman and Simoncelli (2011)* developed a model (hereafter, FS-model) in which texture-like summary statistics are pooled over spatial regions inspired by the receptive fields in primate visual cortex. The size of neural receptive fields in ventral visual stream areas increases as a function of retinal eccentricity, and as one moves downstream from V1 to V2 and V4 at a given eccentricity. Each

visual area therefore has a signature scale factor, defined as the ratio of the receptive field diameter to retinal eccentricity (*Freeman and Simoncelli, 2011*). Similarly, the pooling regions of the FS-model also increase with retinal eccentricity with a definable scale factor. New images can be synthesised that match the summary statistics of original images at this scale factor. As scale factor increases, texture statistics are pooled over increasingly large regions of space, resulting in more distorted synthesised images relative to the original (that is, more information is discarded).

The maximum scale factor for which the images remain indistinguishable (the critical scale) characterises perceptually-relevant compression in the visual system's representation. If the scale factor of the model corresponded to the scaling of the visual system in the responsible visual area, and information in upstream areas was irretrievably lost, then the images synthesised by the model should be indistinguishable while discarding as much information as possible. That is, we seek the maximum compression that is perceptually lossless:

$$s_{\mathrm{crit}}(I) = \max_{s:\,\mathrm{d}(\hat{I}_s, I)=0} s,$$

where $s_{\mathrm{crit}}(I)$ is the critical scale for an image $I$, $\hat{I}_s$ is a synthesised image at scale $s$ and $d$ is a perceptual distance. Larger scale factors discard more information than the relevant visual area and therefore the images should look different. Smaller scale factors preserve information that could be discarded without any perceptual effect.

Crucially, it is the *minimum* critical scale over images that is important for the scaling theory. If the visual system computes summary statistics over fixed (image-independent) pooling regions in the same way as the model, then the model must be able to produce metamers for all images. While images may vary in their individual critical scales, the image with the smallest critical scale determines the maximum compression for appearance to be matched by the visual system in general, assuming an image-independent representation:

$$s_{\mathrm{system}} = \min_{I} s_{\mathrm{crit}}(I)$$

Freeman and Simoncelli showed that the largest scale factor for which two synthesised images could not be told apart was approximately 0.5, or pooling regions of about half the eccentricity. This scaling matched the signature of area V2, and also matched the approximate value of Bouma's Law. Subsequently, this result has been interpreted as demonstrating a link between receptive field scaling, crowding, and our rich phenomenal experience (e.g. *Block, 2013*; *Cohen et al., 2016*, *Landy, 2013*, *Movshon and Simoncelli, 2014*, *Seth, 2014*). These interpretations imply that the FS-model creates metamers for natural scenes. However, observers in Freeman and Simoncelli's experiment never saw the original scenes, but only compared synthesised images to each other. Showing that two model samples are indiscriminable from each other could yield trivial results. For example, two white noise samples matched to the mean and contrast of a natural scene would be easy to discriminate from the scene but hard to discriminate from each other. Furthemore, since synthesised images represent a specific subset of images, and the system critical scale $s_{\mathrm{system}}$ is the minimum over all possible images, the $s_{\mathrm{system}}$ estimated in *Freeman and Simoncelli (2011)* is likely to be an overestimate.

No previous paper has estimated $s_{\mathrm{system}}$ for the FS-model using natural images. *Wallis et al., 2016* tested the related *Portilla and Simoncelli (2000)* model textures, and found that observers could easily discriminate these textures from original images in the periphery. However, the Portilla and Simoncelli model makes no explicit connection to neural receptive field scaling. In addition, relative to the textures tested by *Wallis et al., 2016*, the pooling region overlap used in the FS-model provides a strong constraint on the resulting syntheses, making the images much more similar to the originals. It is therefore still possible that the FS-model produces metamers for natural scenes for scale factors of 0.5.

## Results

### Measuring critical scale in the FS-model

We tested whether the FS-model can produce metamers using an oddity design in which the observer had to pick the odd image out of three successively shown images (*Figure 1E*). In a three-alternative oddity paradigm, performance for metamerism would lie at 1/3 (dashed horizontal line, *Figure 1F*). We used two comparison conditions: either observers compared two model syntheses to each other (synth vs synth; as in *Freeman and Simoncelli, 2011*) or the original image to a model synthesis (orig vs synth). As in the original paper (*Freeman and Simoncelli, 2011*) we measured the performance of human observers for images synthesised with different scale factors (using Freeman and Simoncelli's code, see Materials and methods). To quantify the critical scale factor we fit the same nonlinear model as Freeman and Simoncelli, which parameterises sensitivity as a function of critical scale and gain, but using a mixed-effects model with random effects of participant and image (see Materials and methods).

We used 20 images to test the FS model. These images are split into two classes of ten images each, which we labelled 'scene-like' and 'texture-like'. The distinction of these two classes is based on the results of a pilot experiment with a model we developed, which is inspired by the FS model but based on a different set of image features (those extracted by a convolutional neural network; see Materials and methods and *Appendix 2—figure 1*). In this pilot experiment, we found that some images are easier to discriminate than others (*Appendix 2—figure 7—figure 9*). Easily-discriminable images tended to contain larger areas of inhomogenous structure, long edges, borders between different surfaces or objects, and angled edges providing perspective cues ('scene-like'). Difficult images tended to contain more visual textures: homogenous structure, patterned content, or materials ('texture-like'"). For example, images from the first class tended to contain more structure such as faces, text, skylines, buildings, and clearly segmented objects or people, whereas images from the second class tended to contain larger areas of visual texture such as grass, leaves, gravel, or fur. A similar distinction could also be made along the lines of 'human-made' versus 'natural' image structure, but we suspect the visual structure itself rather than its origin is of causal importance and so used that level of description.

While our labelling of images in this way is debatable (for example, 'texture-like' regions contain some 'scene-like' content and vice versa) and to some degree based on subjective judgment, we hypothesised that this classification distinguishes the types of image content that are critical. If the visual system indeed created a texture-like summary in the periphery and the FS-model was a sufficient approximation of that process, then we should observe no difference in the average critical scale factor of images in each group (because image content would be irrelevant to the distribution of $s_{\mathrm{crit}}(I)$).

We start by considering the condition where participants compared synthesised images to each other—as in *Freeman and Simoncelli (2011)*. Under this condition, there was little evidence that the critical scale depended on the image content (see curves in *Figure 1F*, synth vs synth). The critical scale (posterior mean with 95% credible interval quantiles) for scene-like images was 0.28, 95% CI [0.21, 0.36] and the critical scale for texture-like images was 0.37, 95% CI [0.27, 0.5] (*Figure 1G*). Though these critical scales are lower than those reported by *Freeman and Simoncelli (2011)*, they are within the range of other reported critical scale factors (*Freeman and Simoncelli, 2013*). There was weak evidence for a difference in critical scale between texture-like and scene-like images, with the posterior distribution of scale differences being 0.09, 95% CI [−0.03, 0.24], $\mathrm{p}(\beta < 0) = 0.078$ (where $\mathrm{p}(\beta < 0)$ is the posterior probability of the difference being negative; symmetrical posterior distributions centered on zero would have $\mathrm{p}(\beta < 0) = 0.5$). However, this evidence should be interpreted cautiously: because asymptotic performance never reaches high values, critical scale estimates are more uncertain than in the orig vs synth condition below (*Figure 1G*). This poor asymptotic performance may be because we used more images in our experiment than Freeman and Simoncelli, so participants were less familiar with the distortions that could appear. To make sure this difference did not arise due to different experimental paradigms (oddity vs. ABX), we repeated the experiment using the same ABX task as in Freeman and Simoncelli (*Appendix 1—figure 4*). This experiment again showed poor asymptotic performance, and furthermore demonstrated no evidence for a critical scale difference between the scene- and texture-like images. Taken together, our synth vs synth results are somewhat consistent with Freeman and Simoncelli, who

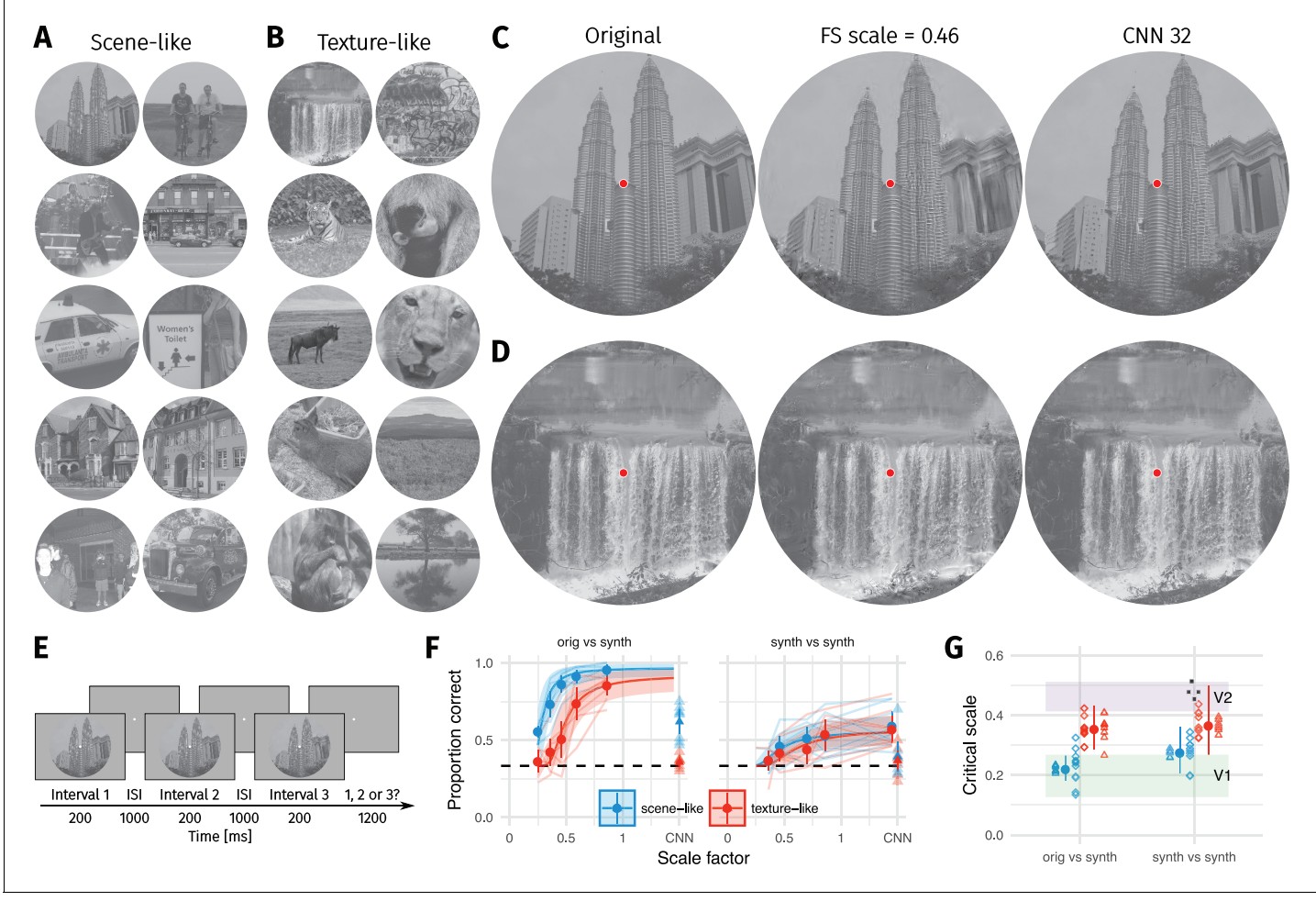

**Figure 1.** Two texture pooling models fail to match arbitrary scene appearance. We selected ten scene-like (**A**) and ten texture-like (**B**) images from the MIT 1003 dataset (*Judd et al., 2009*, https://people.csail.mit.edu/tjudd/WherePeopleLook/index.html) and synthesised images to match them using the Freeman and Simoncelli model (FS scale 0.46 shown) or a model using CNN texture features (CNN 32; example scene and texture-like stimuli shown in (**C**) and (**D**) respectively). Images reproduced under a CC-BY license (https://creativecommons.org/licenses/by/3.0/) with changes as described in the Methods. (**E**): The oddity paradigm. Three images were presented in sequence, with two being physically-identical and one being the oddball. Participants indicated which image was the oddball (1, 2 or 3). On 'orig vs synth' trials participants compared real and synthesised images, whereas on 'synth vs synth' trials participants compared two images synthesised from the same model. (**F**): Performance as a function of scale factor (pooling region diameter divided by eccentricity) in the Freeman-Simoncelli model (circles) and for the CNN 32 model (triangles; arbitrary x-axis location). Points show grand mean ±2 SE over participants; faint lines link individual participant performance levels (FS-model) and faint triangles show individual CNN 32 performance. Solid curves and shaded regions show the fit of a nonlinear mixed-effects model estimating the critical scale and gain. Participants are still above chance for scene-like images in the original vs synth condition for the lowest scale factor of the FS-model we could generate, and for the CNN 32 model, indicating that neither model succeeds in producing metamers. (**G**): When comparing original and synthesised images, estimated critical scales (scale at which performance rises above chance) are lower for scene-like than for texture-like images. Points with error bars show population mean and 95% credible intervals. Triangles show posterior means for participants; diamonds show posterior means for images. Black squares show critical scale estimates of the four participants from *Freeman and Simoncelli (2011)* (x-position jittered to reduce overplotting); shaded regions denote the receptive field scaling of V1 and V2 estimated by *Freeman and Simoncelli (2011)*. Data reproduced from *Freeman and Simoncelli (2011)* using WebPlotDigitizer v. 4.0.0 (Rohatgi, A., software under the GNU Affero General Public License v3, https://www.gnu.org/licenses/agpl-3.0.en.html).

DOI: https://doi.org/10.7554/eLife.42512.003

The following figure supplement is available for figure 1:

**Figure supplement 1.** The ten scene-like and ten texture-like images used in our main experiments, along with example syntheses from the FS-0.46 and CNN 32 models (best viewed with zoom).

DOI: https://doi.org/10.7554/eLife.42512.004

reported no dependency of $s_{\text{crit}(I)}$ on image. It seems likely that this is because comparing synthesised images to each other means that the model has removed higher-order structure that might allow discrimination. All images appear distorted, and the task becomes one of identifying a specific distortion pattern.

Comparing the original image to model syntheses yielded a different pattern of results. First, participants were able to discriminate the original images from their FS-model syntheses at scale factors of 0.5 (*Figure 1F*). Performance lay well above chance for all participants. This result held for both scene-like and texture-like images. Furthermore, there was evidence that critical scale depended on the image type. Model syntheses matched the texture-like images on average with scale factors of 0.36, 95% CI [0.29, 0.43]. In contrast, the scene-like images were quite discriminable from their model syntheses even at the smallest scale we could generate (0.25). The critical scale estimated for scene-like images was 0.22, 95% CI [0.18, 0.27]. Texture-like images had higher critical scales than scene-like images on average (scale difference = 0.13, 95% CI [0.06, 0.22], $p(\beta < 0) = 0.001$).

This difference in critical scale was not attributable to differences in the success of the synthesis procedure between scene-like and texture-like images. Scene-like images had higher final loss (distance between the original and synthesised images in model space) than texture-like images on average (see Materials and methods). This is a corollary of the importance of image content: since a texture summary model is a poor description of scene-like content, the model's optimisation procedure is also more likely to find local minima with relatively high loss. We checked that our main result was not explained by this difference by performing a control analysis in which we refit the model after equating the average loss in the two groups by excluding images with highest final loss until the groups were matched (resulting in four scene-like images being excluded; see Materials and methods). The remaining scene-like images had a critical scale of 0.24, 95% CI [0.2, 0.28] in the orig vs synth condition, texture-like images again showed a critical scale of 0.36, 95% CI [0.3, 0.42] and the difference distribution had a mean of 0.12, 95% CI [0.06, 0.19], $p(\beta < 0) < 0.001$. Thus, differences in synthesis loss do not explain our findings.

As noted above, the image with the minimum critical scale determines the largest compression that can be applied for the scaling model to hold ($s_{\text{system}}$). For two images (*Figure 2A and E*) the nonlinear mixed-effects model estimated critical scales of approximately 0.14 (see *Figure 1G*, diamonds; the minimum critical scale after excluding high-loss images in the control analysis reported above was 0.19). However, examining the individual data for these images (*Figure 2D and H*) reveals that these critical scale estimates are largely determined by the hierarchical nature of the mixed-effects model, not the data itself. Both images were easy to discriminate from the original even for the lowest scale factor we could generate. This suggests that the true scale factor required to generate metamers may be even lower than estimated by the mixed-effects model.

Our results show that smaller pooling regions are required to make metamers for scene-like images than for texture-like images. Human observers can reliably detect relatively small distortions produced by the FS-model at scale factors of 0.25 in scene-like image content (compare *Figure 2B and F* at scale 0.25 and *C* and *G* at scale 0.46 to images *A* and *B*). Thus, syntheses at these scales are not metamers for natural scenes.

## Local image structure determines the visibility of texture-like distortions

In our first experiment we found that scene-like images yielded lower critical scales than texture-like images. However, this categorisation is crude: 'texture-ness' in photographs of natural scenes is a property of local regions of the image rather than the image as a whole. In addition, the classification of images above was based in part on the difficulty of these images in a pilot experiment.

We therefore ran a second experiment to test the importance of local image structure more directly (*Bex, 2010*; *Koenderink et al., 2017*; *Valsecchi et al., 2018*; *Wallis and Bex, 2012*), using a set of images whose selection was not based on pilot discrimination results. Participants detected a localised texture-like distortion (generated by the texture model of *Gatys et al., 2015*) blended into either a scene-like or texture-like region (*Figure 3A–C*). These image regions were classified by author CF (non-authors showed high agreement with this classification—see Materials and methods). The patches were always centered at an eccentricity of six degrees, and we varied the radius of the circular patch (*Figure 3D*). This is loosely analogous to creating summary statistics in a single pooling

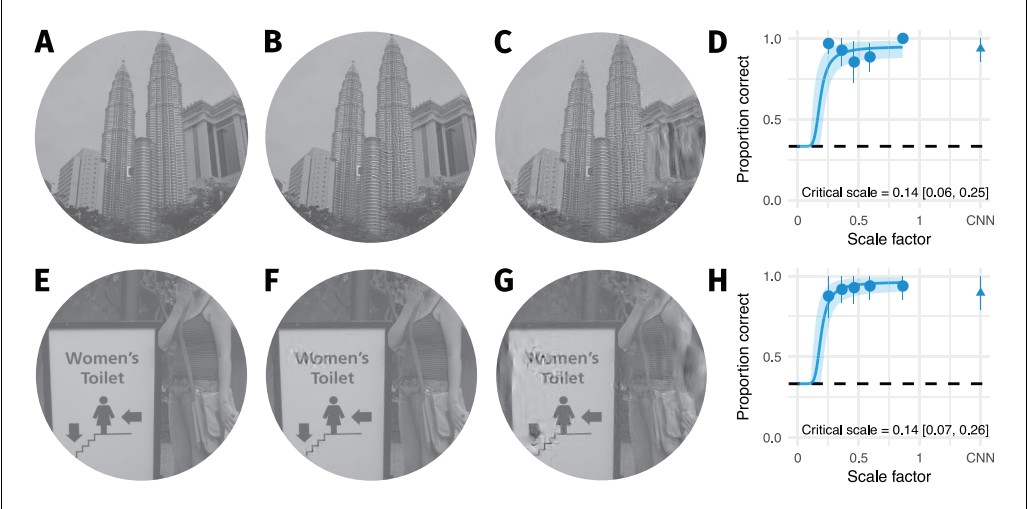

**Figure 2.** The two images with smallest critical scale estimates are highly discriminable even for the lowest scale factor we could generate. (A) The original image. (B) An example FS synthesis at scale factor 0.25. (C) An example FS synthesis at scale factor 0.46. Images in B and C reproduced from the MIT 1003 Database (*Judd et al., 2009*), https://people.csail.mit.edu/tjudd/WherePeopleLook/index.html) under a CC-BY license (https://creativecommons.org/licenses/by/3.0/) with changes as described in the Methods. (D) The average data for this image. Points and error bars show grand mean and ±2 SE over participants, solid curve and shaded area show posterior mean and 95% credible intervals from the mixed-effects model. Embedded text shows posterior mean and 95% credible interval on the critical scale estimate for this image. (E–H) Same as A–D for the image with the second-lowest critical scale. Note that in both cases the model is likely to overestimate critical scale.

DOI: https://doi.org/10.7554/eLife.42512.005

The following figure supplement is available for figure 2:

**Figure supplement 1.** Images with the highest and lowest critical scale estimates within the scene-like and texture-like categories for the orig vs synth comparison.

DOI: https://doi.org/10.7554/eLife.42512.006

region (*Wallis et al., 2016*). Participants discriminated between the original image and an image containing a local distortion in a 2IFC paradigm (*Figure 3E*).

The results showed that the visibility of texture-like distortions depended strongly on the underlying image content. Participants were quite insensitive to even large texture-like distortions occurring in texture-like image regions (*Figure 3F*). Performance for distortions of nearly five degrees radius (i.e. nearly entering the foveal fixation point) was still close to chance. Conversely, distorting scene-like regions is readily detectable for the three largest distortion patch sizes.

## Discussion

It is a popular idea that the appearance of scenes in the periphery is described by summary statistic textures captured at the scaling of V2 neural populations. In contrast, here we show that humans are very sensitive to the difference between original and model-matched images at this scale (*Figure 1*). A recent preprint (*Deza et al., 2017*) finds a similar result in a set of 50 images, and our results are also consistent with the speculations made by Wallis et al. based on their experiments with Portilla and Simoncelli textures (*Wallis et al., 2016*). Together, these results show that the pooling of texture-like features in the FS-model at the scaling of V2 receptive fields does not explain the appearance of natural images.

One exciting aspect of *Freeman and Simoncelli (2011)* was the promise of inferring a critical brain region via a receptive field size prediction derived from psychophysics. Indeed, aspects of this promise have since received empirical support: the presence of texture-like features can discriminate V2 neurons from V1 neurons (*Freeman et al., 2013*; *Ziemba et al., 2016*; see also *Okazawa et al., 2015*). Discarding all higher-order structure not captured by the candidate model by comparing syntheses to each other, thereby isolating only features that change, may be a useful way to distinguish the feedforward component of sequential processing stages in neurons.

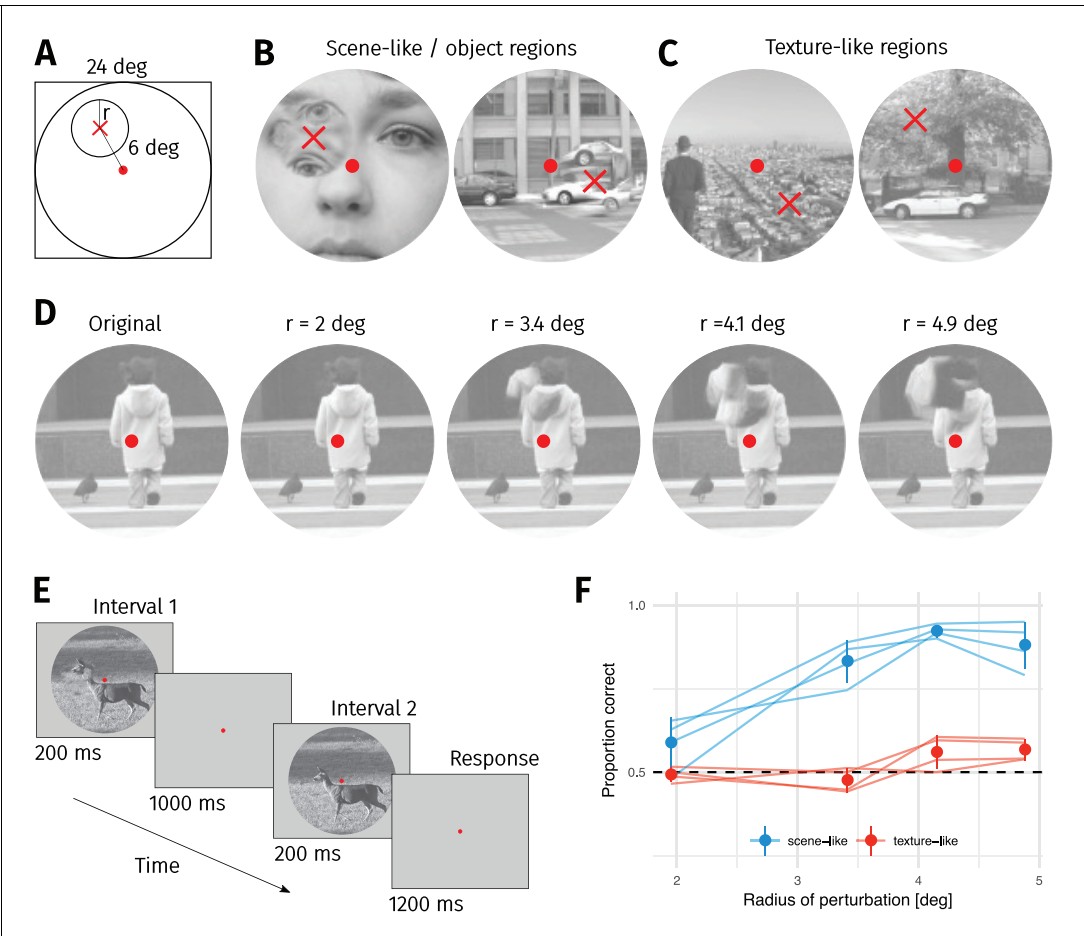

**Figure 3.** Sensitivity to local texture distortions depends on image content. (**A**) A circular patch of an image was replaced with a texture-like distortion. In different experimental conditions the radius of the patch was varied. (**B**) Two example images in which a 'scene-like' or inhomogenous region is distorted (red cross). (**C**) Two example images in which a 'texture-like' or homogenous region is distorted (red cross). (**D**) Examples of an original image and the four distortion sizes used in the experiment. Images in B–D reproduced from the MIT 1003 Database (**Judd et al., 2009**), https://people.csail. mit.edu/tjudd/WherePeopleLook/index.html) under a CC-BY license (https://creativecommons.org/licenses/by/3.0/) with changes as described in the Methods. (**E**) Depiction of the 2IFC task, in which the observer reported whether the first or second image contained the distortion. (**F**) Proportion correct as a function of distortion radius in scene-like (blue) and texture-like (red) image regions. Lines link the performance of each observer (each point based on a median of 51.5 trials; min 31, max 62). Points show mean of observer means, error bars show ±2 SEM.
DOI: https://doi.org/10.7554/eLife.42512.007

While texture-like representations may therefore be important for understanding neural encoding (**Movshon and Simoncelli, 2014**), our results call into question the link between receptive field scaling and scene appearance. If the peripheral appearance of visual scenes is explained by image-independent pooling of texture-like features, then the pooling regions must be small. Consider that participants in our experiment could easily discriminate the images in **Figure 2B and F** from those in **Figure 2A and E** respectively. Therefore, images synthesised at a truly metameric scaling must remain extremely close to the original: $s_{\text{system}}$ must be at least as small as V1 neurons, and perhaps even lower (**Figure 2**). This may even be consistent with scaling in precortical visual areas. For example, the scaling of retinal ganglion cell receptive fields at the average eccentricity of our stimuli (six degrees) is approximately 0.08 for the surround (**Croner and Kaplan, 1995**) and 0.009 for the centre (**Dacey and Petersen, 1992**). It becomes questionable how much is learned about compression in the ventral pathway using such an approach, beyond the aforementioned, relatively well-studied limits of optics and retinal sampling (e.g. **Wandell, 1995**; **Watson, 2014**).

A second main finding from our paper is that the ability of the FS-model to synthesise visual metamers at a given scale factor depends on image content. Images containing predominantly

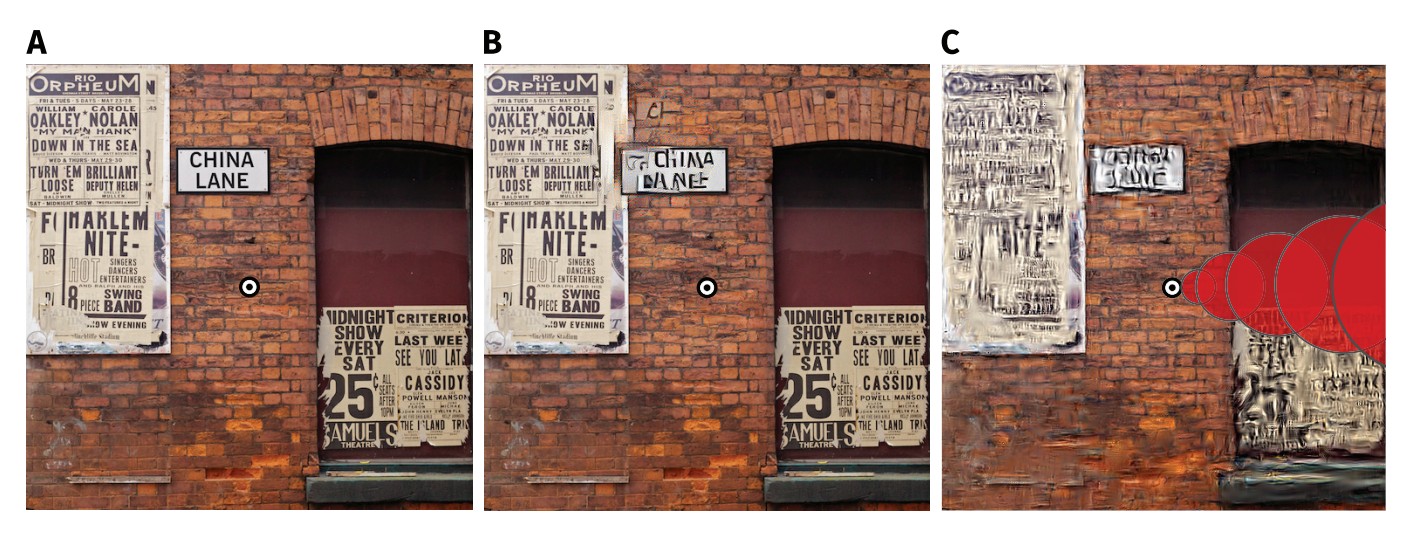

**Figure 4.** The visibility of texture-like distortions depends on image content. (**A**) 'Geotemporal Anomaly' by Pete Birkinshaw (2010: https://www.flickr.com/photos/binaryape/5203086981, re-used under a CC-BY 2.0 license: https://creativecommons.org/licenses/by/2.0/uk/). The image has been resized and a circular bullseye has been added to the centre. (**B**) Two texture-like distortions have been introduced into circular regions of the scene in A (see *Figure 4—figure supplement 1* for higher resolution). The distortion in the upper-left is quite visible, even with central fixation on the bullseye, because it breaks up the high-contrast contours of the text. The second distortion occurs on the brickwork centered on the bullseye, and is more difficult to see (you may not have noticed it until reading this caption). The visibility of texture-like distortions can depend more on image content than on retinal eccentricity (see also *Figure 3*). (**C**) Results synthesised from the FS-model at scale 0.46 for comparison. Pooling regions depicted for one angular meridian as overlapping red circles; real pooling regions are smooth functions tiling the whole image. Pooling in this fashion reduces large distortions compared to B, but our results show that this is insufficient to match appearance.

DOI: https://doi.org/10.7554/eLife.42512.008

The following figure supplement is available for figure 4:

**Figure supplement 1.** Higher-resolution versions of the images from *Figure 4*.

DOI: https://doi.org/10.7554/eLife.42512.009

'scene-like' content tended to be more difficult to match (requiring lower scale factors in the case of the FS-model) than images containing 'texture-like' content (*Figure 1F and G*). In a second experiment measuring the visibility of local texture distortions, we found that people can be quite insensitive to even large texture-like distortions so long as these fall on texture-like regions of the input image (*Figure 3*). This confirms the importance of the distinction between 'things' (scene-like content) and 'stuff' (texture-like content; *Adelson, 2001*) for peripheral scene appearance.

This result can be experienced via simple demonstration. The 'China Lane' sign in *Figure 4A* has been distorted in *Figure 4B* (using local texture distortions as in *Figure 3*), and is readily visible in the periphery (with central fixation on the circular bullseye). The same type of distortion in a texture-like region of the image is far less visible (the brickwork in the image centre; FS-model result *Figure 4C*), despite appearing in the parafovea. It is the image content, not retinal eccentricity, that is the primary determinant of the visibility of at least some summary statistic distortions. Requiring information to be preserved at V1 or smaller scaling would therefore be inefficient from the standpoint of compression: small scale factors will preserve texture-like structure that could be compressed without affecting appearance.

It may seem trivial that a texture statistic model better captures the appearance of textures than non-textures. However, if the human visual system represents the periphery as a texture-like summary, and these models are sufficient approximations of this representation, then image content should not matter—because scene-like retinal inputs in the periphery are transformed into textures by the visual system.

Perhaps the V2 scaling theory holds but the FS-model texture features are insufficient to capture natural scene appearance. To test whether improved texture features (*Gatys et al., 2015*) could help in matching appearance for scenes, we developed a new model (CNN-model; see Materials

and methods and *Appendix 2—figures 1–4*) that was inspired by the FS-model but uses the texture features of a convolutional neural network (VGG-19, *Simonyan and Zisserman, 2015*) that have previously been shown to better capture the appearance of some textures than the Portilla and Simoncelli texture features (*Wallis et al., 2017*). As for the FS-model, discrimination performance becomes poorer as pooling region sizes become smaller (*Appendix 2—figure 3*). The CNN 32 model shows very similar behaviour to the FS-model such that human performance for scene-like images is higher than for texture-like images (triangles in *Figure 1D* and *Figure 2*). Thus, the syntheses from both models are not metamers for natural scenes. Nevertheless, our results cannot rule out that a hereto unknown summary statistic model exists that will create metamers for all images at V2 scales or higher. However, that two additional summary statistic models (the CNN-model and the NeuroFovea model of *Deza et al., 2017*) also fail to capture scene appearance and show dependence on image content adds some generality to our claim that these models are insufficient descriptions of peripheral visual scene appearance.

If this claim was correct, this begs the question: what is the missing ingredient that could capture appearance while compressing as much information as possible? Through the Gestalt tradition, it has long been known that the appearance of local image elements can crucially depend on the context in which they are placed and their interpretation in the scene (for overviews of recent work, see *Jäkel et al., 2016*; *Wagemans et al., 2012a*; *Wagemans et al., 2012b*). We speculate that mechanisms of perceptual organisation (such as segmentation and grouping) need to be considered if one wants to capture appearance in general—yet current models that texturise local regions do not explicitly include these mechanisms (*Herzog et al., 2015*; *Clarke et al., 2014*). If segmentation and grouping processes are critical for efficiently matching scene appearance, then uniformly computing summary statistics without including these processes will require preserving much of the original image structure by making pooling regions very small. A parsimonious model capable of compressing as much information as possible might need to adapt either the size and arrangement of pooling regions or the feature representations to the image content.

## Local vs global mechanisms

These segmentation and grouping mechanisms could be mediated by local interactions between nearby image features, global properties of the scene, or both. The present results do not allow us to distinguish these alternatives.

In favour of the importance of local interactions, studies of contour integration in Gabor fields show that the arrangement of local orientation structure can influence the discrimination of contour shape (*Dakin and Baruch, 2009*) and contour localisation (*Robol et al., 2012*), and that these effects are consistent with crowding (*Robol et al., 2012*). In these stimuli, crowding between nearby contour elements is the primary determinant of global contour judgments (see also *Dakin et al., 2009*). Specifically, contours consisting of parallel Gabor elements ('snakes') were more easily perceived when adjacent Gabor elements were oriented perpendicularly to the main contour. A related study (*Van der Burg et al., 2017*) used an evolutionary algorithm to select dense line element displays that maximally alleviated crowding in an orientation discrimination task. Displays evolved using human responses showed that a substantial reduction of crowding was obtained by orienting the two line segments nearest the target (separated by only $0.75°$ at $6°$ eccentricity) to be perpendicular to the target's mean orientation (forming 'T' and/or 'I' junctions). In contrast, simulations based on Bouma's Law predicted that much larger areas of the display (relative to the human data) would need to be adjusted. These results are consistent with our finding that humans can be far more sensitive to image structure in the periphery than predicted by Bouma-like scaling.

The studies above suggest the possibility that T-junctions may be critical local cues to segmentation in the periphery. The potential importance of different junction types in segmentation and grouping has long been noted (*Biederman, 1987*). In real scenes, T-junctions usually signal occlusion edges between rigid surfaces, whereas Y-, L- and arrow-junctions are created by projecting the corners of 3D objects into 2D. Histograms of junction distributions are diagnostic of scene category (*Walther and Shen, 2014*), with human-made scenes such as city streets and offices tending to contain more T-junctions than more natural environments like beaches and mountains. A recent study also highlights the importance of local contour symmetry for scene categorisation (*Wilder et al., 2019*). Finally, *Loschky et al. (2010)* found that participants were extremely poor at classifying scene category from *Portilla and Simoncelli (2000)* global textures of scene images. These results suggest

that the Portilla and Simoncelli texture statistics (used in the FS-model) do not adequately preserve junction information.

Taken together, these studies give rise to the following hypothesis: images with more junctions (particularly T-junctions; *Van der Burg et al., 2017*) will require smaller pooling regions to match and thus will show lower critical scale estimates in the FS-model. We applied the junction detection algorithm of *Xia et al. (2014)* to each of the 20 original images used in our first experiment. Consistent with the (post-hoc) hypothesis above, lower critical scales were associated with more frequent junctions, particularly if 'less meaningful' junctions (defined by the algorithm) were excluded (T-junction correlation $r = -0.54$; L-junctions $r = -0.63$; *Appendix 1—figure 3*). If confirmed by a targeted experiment (and dissociated from general edge density), this relationship would suggest a clear avenue for future improvement of scene appearance models: they must successfully capture junction information in images.

Other evidence supports the role of global information (the arrangement and organisation of objects over large retinal areas) in segmentation and grouping. In crowding, *Manassi et al. (2013)* found that configurations of stimuli well outside the region of Bouma's law could modulate the crowding effectiveness of the same flankers (see also *Manassi et al., 2012*; *Saarela et al., 2009*; *Vickery et al., 2009*; *Levi and Carney, 2009*). *Neri (2017)* reported evidence from a variety of experiments in support of a fast segmentation process, operating over large regions of space, that can strongly modulate the perceptual interpretation of—and sensitivity to—local edge elements in a scene according to the figure-ground organisation of the scene (see also *Teufel et al., 2018*). Our findings could be explained by the fact that the texture summary statistic models we examine here do not include any such global segmentation processes. The importance of these mechanisms could be examined in future studies, and potentially dissociated from the local information discussed above, by using image manipulations thought to disrupt the activity of global grouping mechanisms such as polarity inversion or image two-toning (*Neri, 2017*; *Balas, 2012*; *Teufel et al., 2018*).

## Summary statistics, performance and phenomenology

Our results do not undermine the considerable empirical support for the periphery-as-summary-statistic theory as a description of visual performance. Humans can judge summary statistics of visual displays (*Ariely, 2001*; *Dakin and Watt, 1997*), summary statistics can influence judgments where other information is lost (*Fischer and Whitney, 2011*; *Faivre et al., 2012*), and the information preserved by summary statistic stimuli may offer an explanation for performance in various visual tasks (*Rosenholtz et al., 2012b*; *Balas et al., 2009*; *Rosenholtz et al., 2012a*; *Keshvari and Rosenholtz, 2016*; *Chang and Rosenholtz, 2016*; *Zhang et al., 2015*; *Whitney et al., 2014*; *Long et al., 2016*; though see *Agaoglu and Chung, 2016*; *Herzog et al., 2015*; *Francis et al., 2017*). Texture-like statistics may even provide the primitives from which form is constructed (*Lettvin, 1976*)—after appropriate segmentation, grouping and organisation. However, one additional point merits further discussion. The studies by Rosenholtz and colleagues primarily test summary statistic representations by showing that performance with summary statistic stimuli viewed foveally is correlated with peripheral performance with real stimuli. This means that the summary statistics preserve sufficient information to explain the performance of tasks in the periphery. Our results show that these summary statistics are insufficient to match scene appearance, at least under the pooling scheme used in the Freeman and Simoncelli model at computationally feasible scales. This shows the usefulness of scene appearance matching as a test: a parsimonious model that matches scene appearance would be expected to also preserve enough information to show correlations with peripheral task performance; the converse does not hold.

While it may be useful to consider summary statistic pooling in accounts of visual performance, to say that summary statistics can account for phenomenological experience of the visual periphery (*Cohen et al., 2016*; see also *Block, 2013*; *Seth, 2014*) seems premature in light of our results (see also *Haun et al., 2017*). *Cohen et al. (2016)* additionally posit that focussed spatial attention can in some cases overcome the limitations imposed by a summary statistic representation. We instead find little evidence that participants' ability to discriminate real from synthesised images is improved by cueing spatial attention, at least in our experimental paradigm and for our CNN-model (*Appendix 2—figure 6*).

## Conclusion

Our results show that the appearance of scenes in the periphery cannot be captured by the *Freeman and Simoncelli (2011)* summary statistic model at receptive field scalings similar to V2. We suggest that peripheral appearance models emphasising pooling processes that depend on retinal eccentricity will instead need to explore input-dependent grouping and segmentation. We speculate that mechanisms of perceptual organisation (either local or global) are critical to explaining visual appearance and efficient peripheral encoding. Models of the visual system that assume image content is processed in feedforward, fixed pooling regions—including current convolutional neural networks—lack these mechanisms.

# Materials and methods

All stimuli, data and code to reproduce the figures and statistics reported in this paper are available at http://dx.doi.org/10.5281/zenodo.1475111. This document was prepared using the knitr package (*Xie, 2013*; *Xie, 2016*) in the R statistical environment (*R Core Team, 2017*; *Wickham and Francois, 2016*; *Wickham, 2009*, *Wickham, 2011*; *Auguie, 2016*; *Arnold, 2016*) to improve its reproducibility.

## Participants

Eight observers participated in the first experiment (*Figure 1*): authors CF and TW, a research assistant unfamiliar with the experimental hypotheses, and five naïve participants recruited from an online advertisement pool who were paid 10 Euro per hr for two one-hour sessions. An additional naïve participant was recruited but showed insufficient eyetracking accuracy (see below). Four observers participated in the second experiment (*Figure 3*); authors CF and TW plus two naïve observers paid 10 Euro per hour. All participants signed a consent form prior to participating. Participants reported normal or corrected-to-normal visual acuity. All procedures conformed to Standard 8 of the American Psychological Association's 'Ethical Principles of Psychologists and Code of Conduct' (2010).

## Stimuli

Images were taken from the MIT 1003 scene dataset (*Judd et al., 2012*; *Judd et al., 2009*). A square was cropped from the center of the original image and downsampled to $512 \times 512$ px. The images were converted to grayscale and standardized to have a mean gray value of 0.5 (scaled [0,1]) and an RMS contrast ($\sigma/\mu$) of 0.3. For the first experiment, images were selected as described in the Results and *Appendix 2—figure 7—figure 9*.

### Freeman and Simoncelli syntheses

We synthesised images using the FS-model (*Freeman and Simoncelli, 2011*, code available from https://github.com/freeman-lab/metamers). Four unique syntheses were created for each source image at each of eight scale factors (0.25, 0.36, 0.46, 0.59, 0.7, 0.86, 1.09, 1.45), using 50 gradient steps as in Freeman and Simoncelli's main experiment. Pilot experiments with stimuli generated with 100 gradient steps produced similar results. *Freeman and Simoncelli (2011)* computed the final loss between original and synthesised images as 'mean squared error, normalized by the parameter variance'. We take this to mean the following: for a matrix of model parameters from an original image $X_{orig}$ (rows are parameters and columns are pooling regions) and the corresponding parameters for the synthesised image $X_{synth}$, we compute the normalised MSE as $MSE = \text{mean}((X_{orig} - X_{synth})^2)/\text{Var}(X_{orig})$. Freeman and Simoncelli report that this metric was $0.01 \pm 0.015$ (mean ± s.d.) across all images and scales in their experiment. For our experiment, the same metric across all images and scales was $0.06 \pm 0.2$. These higher final loss values were driven by the scene-like images, which had a mean loss of $0.11 \pm 0.27$ compared to the texture-like images ($0.01 \pm 0.05$). Excluding the four highest-loss images (all scene-like) reduced the average loss of the scene-like category to $0.01 \pm 0.02$, which is similar to the range of the syntheses used by *Freeman and Simoncelli (2011)* and to the texture-like images. A control analysis showed the difference in critical scale between the image categories remained after matching the average loss (Results).

To successfully synthesise images at scale factors of 0.25 and 0.36 it was necessary to increase the central region of the image in which the original pixels were perfectly preserved (pooling regions near the fovea become too small to compute correlation matrices). Scales of 0.25 used a central radius of 32 px (0.8 dva in our viewing conditions) and scales 0.36 used 16 px (0.4 dva). This change should, if anything, make syntheses even harder to discriminate from the original image. All other parameters of the model were as in Freeman and Simoncelli. Synthesising an image with scale factor 0.25 took approximately 35 hr, making a larger set of syntheses or source images infeasible. It was not possible to reliably generate images with scale factors lower than 0.25 using the code above.

## CNN model syntheses

The CNN pooling model (triangles in *Figure 1*) was inspired by the model of Freeman and Simoncelli, with two primary differences: first, we replaced the *Portilla and Simoncelli (2000)* texture features with the texture features derived from a convolutional neural network (*Gatys et al., 2015*), and second, we simplified the 'foveated' pooling scheme for computational reasons. Specifically, for the CNN 32 model presented above, the image was divided up into 32 angular regions and 28 radial regions, spanning the outer border of the image and an inner radius of 64 px. Within each of these regions we computed the mean activation of the feature maps from a subset of the VGG-19 network layers (conv1_1, conv2_1, conv3_1, conv4_1, conv5_1). To better capture long-range correlations in image structure, we computed these radial and angular regions over three spatial scales, by computing three networks over input sizes 128, 256 and 512 px. Using this multiscale radial and angular pooling representation of an image, we synthesised new images to match the representation of the original image via iterative gradient descent (*Gatys et al., 2015*). Specifically, we minimised the mean-squared distance between the original and a target image, starting from Gaussian noise outside the central 64 px region, using the L-BFGS optimiser as implemented in scipy (*Jones et al., 2001*) for 1000 gradient steps, which we found in pilot experiments was sufficient to produce small (but not zero) loss. Further details, including tests of other variants of this model, are provided in Appendix 2.

## Local distortion experiment

We identified local regions that were scene-like or texture-like, whose centre-of-mass was approximately 128 px (±5 px; approximately 6 degrees) from the centre of the image. Because we are not aware of any algorithmic method to distinguish these types of image structure, these were chosen based on our definition of scene-like and texture-like image content (see Results) by author CF. Specifically, a Python script was used to display the 1003 images of the MIT database with a circle of radius 128 px superimposed. CF clicked on a point on the circle that lay in a texture- or scene-like region; if no such region was identified this image was discarded. The coordinates of this point as well as its classification were stored. This procedure resulted in 389 unique images, of which 229 contained a 'scene-like' region and 160 contained a 'texture-like' region.

Non-authors generally agreed with this classification. We conducted a pilot experiment to measure agreement in five participants. Participants were shown each of the 389 images above with a circle (of radius 100 px) superimposed over the region defined by CF. They were instructed to classify the circled region as 'scene-like' (defined as 'tend to contain larger areas of inhomogenous structure, long edges, borders between different surfaces or objects, and angled edges providing perspective cues') or 'texture-like' (defined as 'homogenous structure, patterned content, or materials') in a single-interval binary response task. We found a mean agreement of 88.6% with CF's classification (individual accuracies of 74.8, 90.2, 92.5, 92.8, 92.8%, mean $d'$ = 2.81, with a mean bias to respond 'scene-like', $\log \beta = -1.39$). In this experiment (conducted approximately two years after the initial classification), CF showed a retest agreement of 97.4%.

For each image we perturbed a circular patch in the center of the texture/object region using the texture model of *Gatys et al. (2015)*. Note that this is the texture model not the CNN-model using radial and angular pooling regions. For each original image, we generated new images containing distortions of different sizes (radii of 40, 70, 85 and 100 px, corresponding to approximately 2, 3.4, 4.1 and 4.9 dva). The local texture features were computed as the (square) Gram matrices in the same VGG-19 layers as used in the CNN-model over an area equal to the radius plus 24 px (square side length $2(r + 24)$). Texture synthesis was then performed via gradient descent as in the CNN-

model, with the exception that the loss function included a circular cosine spatial windowing function which ramped between the synthesised and original pixels over a region of 12 px, in order to smoothly blend the texture distortion with the surrounding image structure. Some example images are shown in *Figure 3*. In total we therefore used 389 unique images and 389*4 synthesised images as stimuli in this experiment.

## Equipment

Stimuli were displayed on a VIEWPixx 3D LCD (VPIXX Technologies Inc, Saint-Bruno-de-Montarville, Canada; spatial resolution 1920 × 1080 pixels, temporal resolution 120 Hz, operating with the scanning backlight turned off in normal colour mode). Outside the stimulus image the monitor was set to mean grey. Participants viewed the display from 57 cm (maintained via a chinrest) in a darkened chamber. At this distance, pixels subtended approximately 0.025 degrees on average (approximately 40 pixels per degree of visual angle). The monitor was linearised (maximum luminance 260 $cd/m^2$) using a Konica-Minolta LS-100 (Konica-Minolta Inc, Tokyo, Japan). Stimulus presentation and data collection was controlled via a desktop computer (Intel Core i5-4460 CPU, AMD Radeon R9 380 GPU) running Ubuntu Linux (16.04 LTS), using the Psychtoolbox Library (version 3.0.12, *Brainard, 1997*; *Kleiner et al., 2007*; *Pelli, 1997*), the Eyelink toolbox (*Cornelissen et al., 2002*) and our internal iShow library (http://dx.doi.org/10.5281/zenodo.34217) under MATLAB (The Mathworks Inc, Natick MA, USA; R2015b). Participants' gaze position was monitored by an Eyelink 1000 (SR Research) video-based eyetracker.

## Procedure

In the first experiment, participants were shown three images in succession on each trial. Two images were identical, one image was different (the 'oddball', which could occur first, second or third with equal probability). The oddball could be either a synthesised or a natural image (in the orig vs synth condition; counterbalanced), whereas the other two images were physically the same as each other and from the opposite class as the oddball. In the synth vs synth condition (as used in Freeman and Simoncelli), both oddball and foil images were (physically different) model synths. The participant identified the temporal position of the oddball image via button press. Participants were told to fixate on a central point (*Thaler et al., 2013*) presented in the center of the screen. The images were centred around this spot and displayed with a radius of 512 pixels (i.e. images were upsampled by a factor of two for display), subtending ≈12.8° at the eye. Images were windowed by a circular cosine, ramping the contrast to zero in the space of 52 pixels. The stimuli were presented for 200 ms, with an inter-stimulus interval of 1000 ms (making it unlikely participants could use motion cues to detect changes), followed by a 1200 ms response window. Feedback was provided by a 100 ms change in fixation cross brightness. Gaze position was recorded during the trial. If the participant moved the eye more than 1.5 degrees away from the fixation spot, the trial immediately ended and no response was recorded; participants saw a feedback signal (sad face image) indicating a fixation break. Prior to the next trial, the state of the participant's eye position was monitored for 50 ms; if the eye position was reported as more than 1.5 degrees away from the fixation spot a recalibration was triggered. The inter-trial interval was 400 ms.

Scene-like and texture-like images were compared under two comparison conditions (orig vs synth and synth vs synth; see main text). Image types and scale factors were randomly interleaved within a block of trials (with a minimum of one trial from another image in between) whereas comparison condition was blocked. Participants first practiced the task and fixation control in the orig vs synth comparison condition (scales 0.7, 0.86 and 1.45); the same images used in the experiment were also used in practice to familiarise participants with the images. Participants performed at least 60 practice trials, and were required to achieve at least 50% correct responses and fewer than 20% fixation breaks before proceeding (as noted above, one participant failed). Following successful practice, participants performed one block of orig vs synth trials, which consisted of five FS-model scale factors (0.25, 0.36, 0.46, 0.59, 0.86) plus the CNN 32 model, repeated once for each image to give a total of 120 trials. The participant then practiced the synth vs synth condition for at least one block (30 trials), before continuing to a normal synth vs synth block (120 trials; scale factors of 0.36, 0.46, 0.7, 0.86, 1.45). Over two one-hour sessions, naïve participants completed a total of four

blocks of each comparison condition in alternating order (except for one participant who ran out of time to complete the final block). Authors performed more blocks (total 11).

In the second experiment, observers discriminated which image contained the distortion in a 2IFC paradigm. Each image was presented for 200 ms with a 1000 ms inter-stimulus interval, after which the observer had 1200 ms to respond. The original, unmodified image could appear either first or second; the other image was the same but contained the circular distortion. Observers fixated a spot (*Thaler et al., 2013*) in the centre of the screen. Feedback was provided, and eyetracking was not used. All observers performed 389 trials. To avoid effects of familiarity with the distortion region, each observer saw each original image only once (that is, each original image was randomly assigned to one of the four distortion scales for each observer). While authors were familiar with the images, naïve observers were not. The consistency of effects between authors and naïves suggests that familiarity does not play a major role in this experiment.

### Data analysis

In the first experiment, we discarded trials for which participants made no response (N = 66) and broke fixation (N = 239), leaving a total of 7555 trials for further analysis. The median number of responses for each image at each scale for each subject in each condition was 4 trials (min 1, max 7). The individual observer data for the FS-model averaged over images (faint lines in *Figure 1F*) were based on a median of 39 trials (min 20, max 70) for each scale in each condition. The individual observer performance as a function of condition (each psychometric function of FS-scale) was based on a median of 192.5 responses (min 136, max 290).

In the second experiment we discarded trials with no response (N = 8), and did not record eye movements, leaving 1548 trials for further analysis.

To quantify the critical scale as a function of the scale factor $s$, we used the same 2-parameter function for discriminability $d'$ fitted by Freeman and Simoncelli:

$$d'(s) = \begin{cases} \alpha\left(1 - \frac{s_c^2}{s^2}\right), & s > s_c \\ 0, & s \leq s_c \end{cases}$$

consisting of the critical scale $s_c$ (below which the participant cannot discriminate the stimuli) and a gain parameter $\alpha$ (asymptotic performance level in units of $d'$). This $d'$ value was transformed to proportion correct using a Weibull function as in *Wallis et al., 2016*:

$$\mathrm{p(correct)} = \frac{1}{m} + \left(1 - \frac{1}{m}\right)\left(1 - \exp\left(-(d'/\lambda)^k\right)\right)$$

with $m$ set to three (the number of alternatives), and scale $\lambda$ and shape $k$ parameters chosen by minimising the squared difference between the Weibull and simulated results for oddity as in *Craven (1992)*. The posterior distribution over model parameters ($s_c$ and $\alpha$) was estimated in a nonlinear mixed-effects model with fixed effects for the experimental conditions (comparison and image type) and random effects for participant (crossed with comparison and image type) and image (crossed with comparison, nested within image type), assuming binomial variability. Note that $s_c$ here is shorthand for a population-level critical scale and group-level offsets estimated for each participant and image; $s_{\mathrm{crit}}(I)$ is the image-specific $s_c$ estimate. Estimates were obtained by a Markov Chain Monte Carlo (MCMC) procedure implemented in the Stan language (version 2.16.2, *Stan Development Team, 2017*; *Hoffman and Gelman, 2014*), with the model wrapper package brms (version 1.10.2, *Bürkner, 2017*; *Bürkner, 2018*) in the R statistical environment. MCMC sampling was conducted with four chains, each with 20,000 iterations (10,000 warmup), resulting in 40,000 post-warmup samples in total. Convergence was assessed using the $\hat{R}$ statistic (*Brooks and Gelman, 1998*) and by examining traceplots. The model parameters were given weakly-informative prior distributions, which provide information about the plausible scale of parameters but do not bias the direction of inference. Specifically, both critical scale and gain were estimated on the natural logarithmic scale; the mean log critical scale (intercept) was given a Gaussian distribution prior with mean −0.69 (corresponding to a critical scale of approximately 0.5—that is centred on the result from Freeman and Simoncelli) and sd 1, other fixed-effect coefficients were given Gaussian priors with mean 0 and sd 0.5, and the group-level standard deviation parameters were given positive-truncated Cauchy priors with mean 0 and sd 0.1. Priors for the log gain parameter were the same,

except the intercept prior had mean 1 (linear gain estimate of 2.72 in $d'$ units) and sd 1. The posterior distribution represents the model's beliefs about the parameters given the priors and data. This distribution is summarised above as posterior mean, 95% credible intervals and posterior probabilities for the fixed-effects parameters to be negative (the latter computed via the empirical cumulative distribution of the relevant MCMC samples).

## Acknowledgments

Funded by the German Federal Ministry of Education and Research (BMBF) through the Bernstein Computational Neuroscience Program Tübingen (FKZ: 01GQ1002), the German Excellency Initiative through the Centre for Integrative Neuroscience Tübingen (EXC307), and the Deutsche Forschungsgemeinschaft (DFG; priority program 1527, BE 3848/2–1 and Projektnummer 276693517 – SFB 1233, Robust Vision: Inference Principles and Neural Mechanisms, TP03). We acknowledge support by the Deutsche Forschungsgemeinschaft and Open Access Publishing Fund of University of Tübingen. We thank Wiebke Ringels for assistance with data collection, Heiko Schütt, Matthias Kümmerer and Corey Ziemba for helpful comments on an earlier draft, Andrew Haun and Ben Balas for helpful comments on Twitter, and reviewer John Cass for suggesting the importance of junction information in explaining our results. TSAW was supported in part by an Alexander von Humboldt Postdoctoral Fellowship. The authors thank the International Max Planck Research School for Intelligent Systems (IMPRS-IS) for supporting Christina Funke.

## Additional information

### Competing interests

Leon A Gatys: This author now works for Apple, Inc. The author's contributions to this article were prior to commencing employment at Apple. The other authors declare that no competing interests exist.

### Funding

| Funder | Grant reference number | Author |
|---|---|---|
| Bundesministerium für Bildung und Forschung | FKZ: 01GQ1002 | Felix A Wichmann<br>Matthias Bethge |
| Deutsche Forschungsgemeinschaft | BE 3848/2-1 | Matthias Bethge |
| Deutsche Forschungsgemeinschaft | Priority program 1527 | Felix A Wichmann<br>Matthias Bethge |
| Deutsche Forschungsgemeinschaft | 276693517; SFB 1233 | Thomas SA Wallis<br>Christina M Funke<br>Felix A Wichmann<br>Matthias Bethge |
| Alexander von Humboldt-Stiftung | Thomas Wallis | Thomas SA Wallis |

The funders had no role in study design, data collection and interpretation, or the decision to submit the work for publication.

### Author contributions

Thomas SA Wallis, Conceptualization, Resources, Data curation, Software, Formal analysis, Supervision, Funding acquisition, Validation, Investigation, Visualization, Methodology, Writing—original draft, Project administration, Writing—review and editing; Christina M Funke, Conceptualization, Resources, Software, Formal analysis, Validation, Investigation, Methodology, Writing—original draft, Writing—review and editing; Alexander S Ecker, Conceptualization, Supervision, Methodology, Project administration, Writing—review and editing; Leon A Gatys, Resources, Software, Methodology, Writing—review and editing; Felix A Wichmann, Conceptualization, Supervision, Funding acquisition, Methodology, Project administration, Writing—review and editing; Matthias Bethge,

Conceptualization, Supervision, Funding acquisition, Project administration, Writing—review and editing

### Author ORCIDs
Thomas SA Wallis https://orcid.org/0000-0001-7431-4852
Alexander S Ecker http://orcid.org/0000-0003-2392-5105
Felix A Wichmann https://orcid.org/0000-0002-2592-634X
Matthias Bethge http://orcid.org/0000-0002-6417-7812

### Ethics

Human subjects: All participants provided informed consent to participate in the study and for their anonymised data to be made publicly available. The study adhered to Standard 8 of the American Psychological Association's "Ethical Principles of Psychologists and Code of Conduct" (2010). The experiments were approved by the Ethics Commission of the University Clinics Tübingen (Nr. 222/2011B02).

### Decision letter and Author response

Decision letter https://doi.org/10.7554/eLife.42512.030
Author response https://doi.org/10.7554/eLife.42512.031

---

## Additional files

### Supplementary files

• Transparent reporting form
DOI: https://doi.org/10.7554/eLife.42512.010

### Data availability

All raw data, processed data, model files, stimulus materials, and analysis code are provided for download in a Zenodo database at http://dx.doi.org/10.5281/zenodo.1475111.

The following dataset was generated:

| Author(s) | Year | Dataset title | Dataset URL | Database and Identifier |
|---|---|---|---|---|
| Wallis TSA, Funke CM | 2018 | Materials to reproduce Wallis, Funke et al. "Image content is more important than Bouma's Law for scene metamers" | http://dx.doi.org/10.5281/zenodo.1475111 | Zenodo, 10.5281/zenodo.1475111 |

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

## Appendix 1

DOI: https://doi.org/10.7554/eLife.42512.011

# Additional experiments with the Freeman and Simoncelli model

## Stimulus artifact control

During the course of our testing we noticed that synthesised images generated with the code from http://github.com/freeman-lab/metamers contained an artifact, visible as a wedge in the lower-left quadrant of the synthesised images in which the phases of the surrounding image structure were incorrectly matched (*Appendix 1—figure 1A*). The angle and extent of the wedge changed with the scale factor, and corresponded to the region where angular pooling regions wrapped from 0–2π (*Appendix 1—figure 1B–C*). The visibility of the artifact depended on image structure, but was definitely due to the synthesis procedure itself because it also occurred when synthesising matches to a white noise source image (*Appendix 1—figure 1D–E*). The artifact was not peculiar to our hardware or implementation because it is also visible in the stimuli shown in *Deza et al. (2017)*.

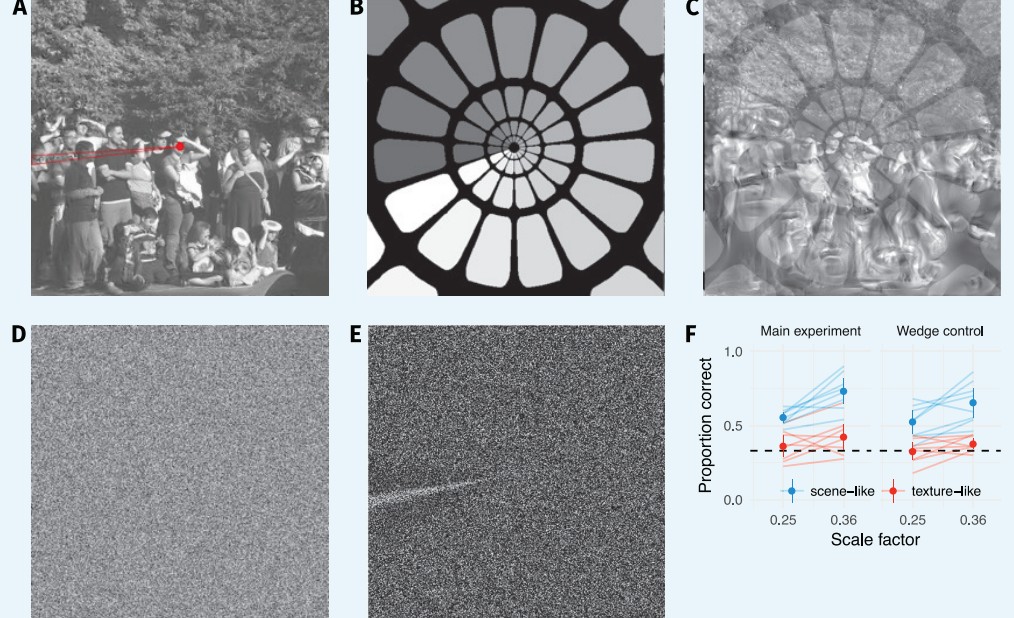

**Appendix 1—figure 1.** Our results do not depend on an artifact in the synthesis procedure. (**A**) During our pilot testing, we noticed a wedge-like artifact in the synthesis procedure of Freeman and Simoncelli (highlighted in red wedge; image from https://github.com/freeman-lab/metamers and shared under a CC-BY license (https://creativecommons.org/licenses/by/3.0/)). The artifact occurred where the angular pooling regions wrapped from 0 to 2π (**B**) pooling region contours shown with increasing greyscale to wrap point, (**C**) overlayed on scene with artifact. (**D**) The artifact was not driven by image content, because it also occurred when synthesising to match white noise (shown with enhanced contrast in (**E**)). If participants' good performance at small scale factors was due to taking advantage of this wedge, removing it by masking out that image region should drop performance to chance. (**F**) Performance at the two smallest scale factors replotted from the main experiment (left) and with a wedge mask overlayed (right) in the orig vs synth comparison. Points show average (±2SE) over participants; faint lines link individual participant means. Performance remains above chance

for the scene-like images, indicating that the low critical scales we observed were not due to the wedge artifact.

DOI: https://doi.org/10.7554/eLife.42512.012

Participants in our experiment could have learned to use the artifact to help discriminate images, particularly synthesised images from original images (since only synthesised images contain the artifact). This may have boosted their sensitivity more than might be expected from the model described by Freeman and Simoncelli, leading to the lower critical scales we observed. To control for this, we re-ran the original vs synth condition with the same participants, with the exception that the lower-left quadrant of the image containing the artifact was masked by a grey wedge (in both original and synthesised images) with angular subtense of 60 degrees. We used only the lowest two scale factors from the main experiment, and participants completed this control experiment after the main experiment reported in the paper. We discarded trials for which participants made no response (N = 9) or broke fixation (N = 57), leaving a total of 1014 trials for further analysis. If the high sensitivity at low scale factors we observed above were due to participants using the artifact, then their performance with the masked stimuli should fall to chance for low scale factors.

This is not what we observed: while performance with the wedge was slightly worse (perhaps because a sizable section of the image was masked), the scene-like images remained above chance performance for the lowest two scale factors (*Appendix 1—figure 1F*). This shows that the low critical scale factors we observed in the main experiment are not due to the wedge artifact.

We performed one additional artifact control experiment. The FS algorithm preserves a small central region of the image exactly in order to match foveal appearance. If there is any image artifact produced by the synthesis procedure at the border of this region, participants could have used this artifact to discriminate the stimuli in the original vs synth condition. Authors TW and CF performed new oddity discrimination trials in which a grey annular occluding zone (inner radius 0.4 deg, outer radius 1.95 deg) was presented over all images. If the low scale factors we find are because participants used a stimulus artifact, then performance at the low scales should drop to chance.

The results of this additional experiment are shown in Figure (*Appendix 1—figure 2*). Both participants can still discriminate real and synthesised scene-like images better than chance even after superposition of the occluding annulus, indicating that any central artifact is not a crucial determinant of discriminability.

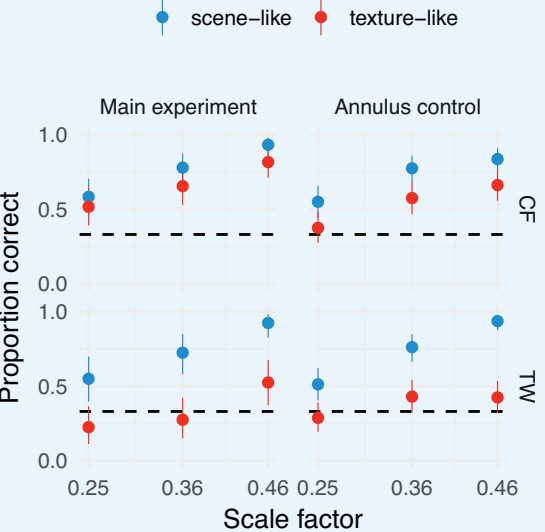

**Appendix 1—figure 2.** Our results do not depend on any potential annular artifact resulting from the synthesis procedure. Performance at the three smallest scale factors replotted from the main experiment (left) and with an annular mask overlayed (right) in the orig vs synth

comparison for authors TW and CF. Points show average performance (error bars show 95% beta distribution confidence limits). Performance remains above chance for the scene-like images, indicating that the low critical scales we observed were not due to a potential annular artifact.

DOI: https://doi.org/10.7554/eLife.42512.013

## Junctions in original images

For each of the 20 original images used in our first experiment, we used the junction detection algorithm of *Xia et al. (2014)* to identify junctions in the image (with algorithm parameter $r_{max} = 36$). We subdivided all three-edge junctions into T-, Y- and arrow-junctions according to the angle criteria used in *Walther and Shen (2014)*, and excluded all junctions that fell outside the circular region of the image shown in our experiment.

We find that scene-like images tend to contain more junctions than texture-like images (*Appendix 1—figure 3A*). This relationship became stronger when we excluded 'less meaningful' junctions (using a 'meaningfulness' cutoff of $\log(\text{NFA}) = -20$, *Xia et al. (2014)*; *Appendix 1—figure 3C*). Images with smaller critical scales are associated with the presence of junctions (*Appendix 1—figure 3B*), and this association gets stronger when small and weak junctions are excluded (*Appendix 1—figure 3D*).

If junction information is important for scene appearance and the FS-model fails to adequately capture this information, we would expect such a negative relationship between junctions and critical scales. Of course, the analysis above does not support a specific causal role for junction information: for example, it may be correlated with simple edge density. Future studies could confirm (or reject) this relationship using a larger and more diagnostic image set.

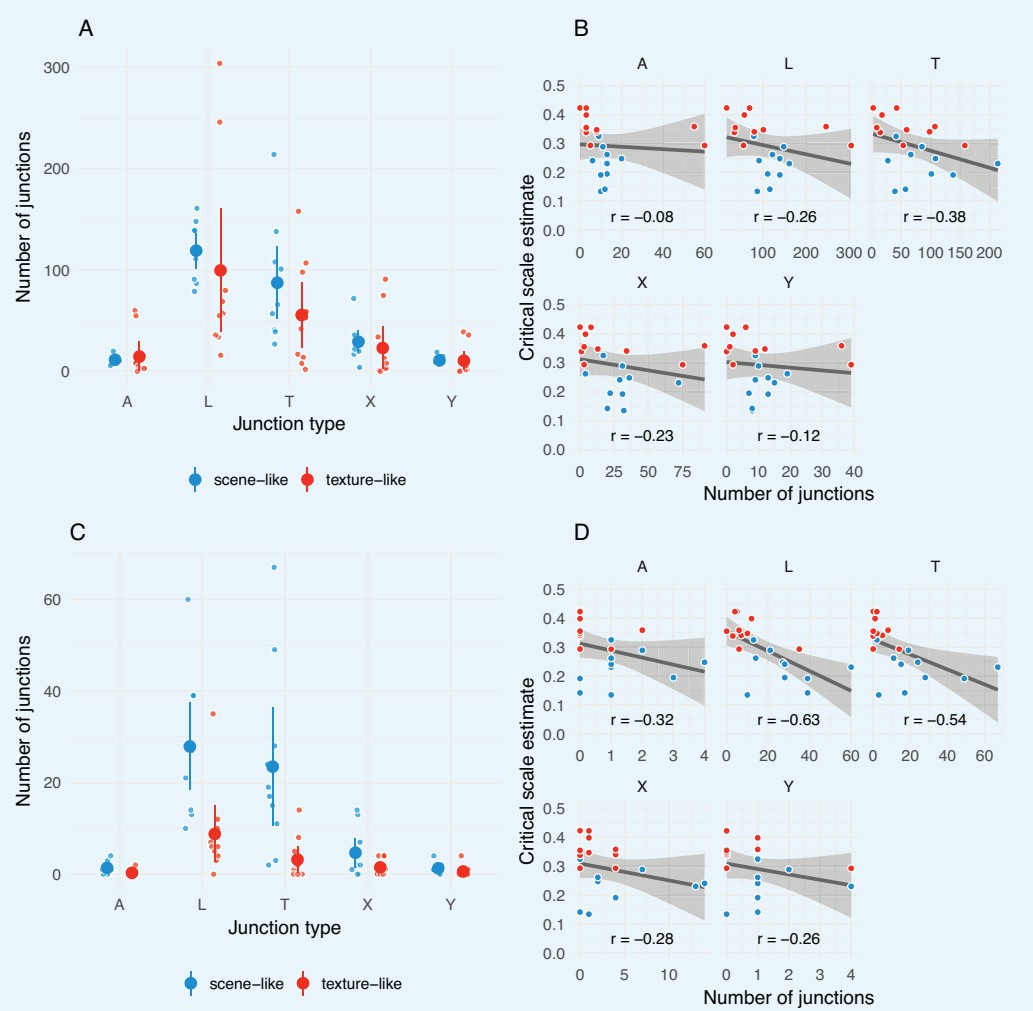

**Appendix 1—figure 3.** The number of junctions present in original images may be related to critical scale estimates. (**A**) Distribution of arrow (A), L-, T-, X- and Y-junctions at all scales and all levels of 'meaningfulness' (*Xia et al., 2014*) in scene-like and texture-like images. Each small point is one image; larger points with error bars show mean ±2 SE. Points have been jittered to aid visibility. (**B**) Correlations between number of junctions of each type with critical scale estimates for that image from the main experiment. Grey line shows linear model fit with shaded region showing 95% confidence area. Pearson correlation coefficient shown below. Note that the x-axis scales in the subplots differ. (**C**) Same as A but for junctions defined with a more strict 'meaningfulness' cutoff of $\log(\text{NFA}) = -20$ (*Xia et al., 2014*). (**D**) Same as B for more 'meaningful' junctions as in C.

DOI: https://doi.org/10.7554/eLife.42512.014

## ABX replication

Participants in our experiment showed poor performance in the synth vs synth condition even for large scale factors (highest accuracy for a participant at the largest scale of 1.45 was 0.8, average accuracy 0.58), leading to relatively flat psychometric functions (*Figure 2F* of main manuscript). In contrast, most participants in *Freeman and Simoncelli (2011)* achieved accuracies above 90% correct for the highest scale factor they test (1.45 as in our experiment). One difference between our experiment and *Freeman and Simoncelli (2011)* is that they used an ABX task, in which participants saw two images A and B, followed by image X, and had to report whether image X was the same as A or B. Perhaps our oddity task is simply harder: due to greater memory load or the cognitive demands of the comparison, participants in our experiment were unable to perform consistently well.

To assess whether the use of an oddity task lead to our finding of lower critical scales and/or poorer asymptotic performance in the synth vs synth condition, we re-ran our experiment as an ABX task. We used the same timing parameters as in Freeman and Simoncelli. Six participants participated in the experiment, including a research assistant (the same as in the main experiment), four naÃve participants and author AE (who only participated in the synth vs synth condition). We discarded trials for which participants made no response (N = 61) or broke fixation (N = 442), leaving a total of 7537 trials for further analysis. The predicted proportion correct in the ABX task was derived from $d'$ using the link function given by **Macmillan and Creelman (2005)**, (229–33) for a differencing model in a roving design:

$$\text{p(correct)} = \Phi\left(\frac{d'(s)}{\sqrt{2}}\right)\Phi\left(\frac{d'(s)}{\sqrt{6}}\right) + \Phi\left(\frac{-d'(s)}{\sqrt{2}}\right)\Phi\left(\frac{-d'(s)}{\sqrt{6}}\right)$$

where $\Phi$ is the standard cumulative Normal distribution.

As in our main experiment with the oddity task, we find that participants could easily discriminate scene-like syntheses from their original at all scales we could generate (**Appendix 1—figure 4**). Critical scale factor estimates were similar to those in the main experiment, indicating that the ABX task did not make a large difference to these results. Critical scale estimates were slightly larger, but much more uncertain, in the synth vs synth condition. This uncertainty is largely driven by the even poorer asymptotic performance than in the main experiment. This shows that the results we report in the primary manuscript are not particular to the oddity task.

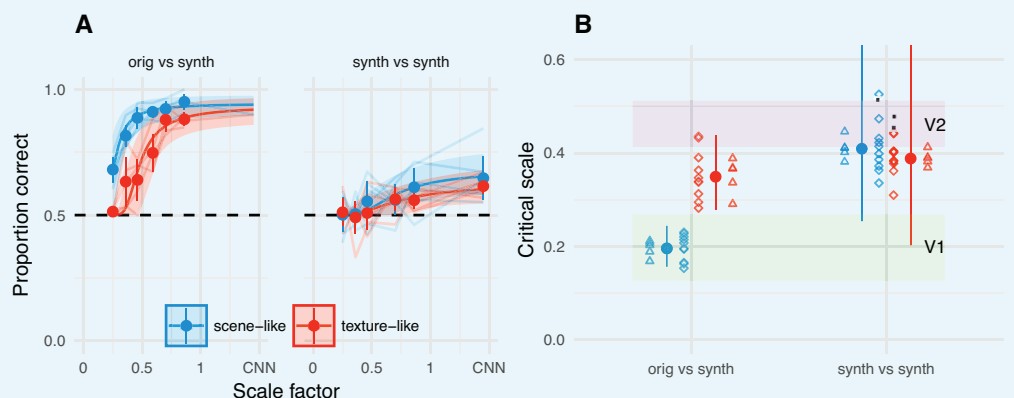

**Appendix 1—figure 4.** Results from the main paper replicated under an ABX task. (**A**) Performance in the ABX task as a function of scale factor. Points show grand mean ±2 SE over participants; faint lines link individual participant performance levels. Solid curves and shaded regions show the fit of a nonlinear mixed-effects model estimating the critical scale and gain. (**B**) When comparing original and synthesised images, estimated critical scales (scale at which performance rises above chance) are lower for scene-like than for texture-like images. Points with error bars show population mean and 95% credible intervals. Triangles show posterior means for participants; diamonds show posterior means for images. Black squares show critical scale estimates of the four participants from Freeman and Simoncelli reproduced from that paper (x-position jittered to reduce overplotting); shaded regions denote the receptive field scaling of V1 and V2 estimated by Freeman and Simoncelli.
DOI: https://doi.org/10.7554/eLife.42512.015

What explains the discrepancy between asymptotic performance in our experiment vs Freeman and Simoncelli? One possibility is that the participants in Freeman and Simoncelli's experiment were more familiar with the images shown, and that good asymptotic performance in the synth vs synth condition requires strong familiarity. Freeman and Simoncelli used four original (source) images, and generated three unique synthesised images for each source image at each scale, compared to our 20 source images with four syntheses.

## Appendix 2

DOI: https://doi.org/10.7554/eLife.42512.011

## CNN scene appearance model

Here we describe the CNN scene appearance model presented in the paper in more detail, as well as additional experiments concerning this model.

To create a summary statistic model using CNN features, we compute the mean activation in a subset of CNN layers over a number of radial and angular spatial regions (see *Appendix 2—figure 1*). Increasing the number of pooling regions (reducing the spatial area over which CNN features are pooled) preserves more of the structure of the original image. New images can be synthesised by minimising the difference between the model features for a given input image and a white noise image via an iterative gradient descent procedure (see below). This allows us to synthesise images that are physically different to the original but approximately the same according to the model. We did this for each of four pooling region sizes, named model 4, 8, 16 and 32 respectively after the number of angular pooling regions. These features were matched over three spatial scales, which we found improved the model's ability to capture long-range correlations.

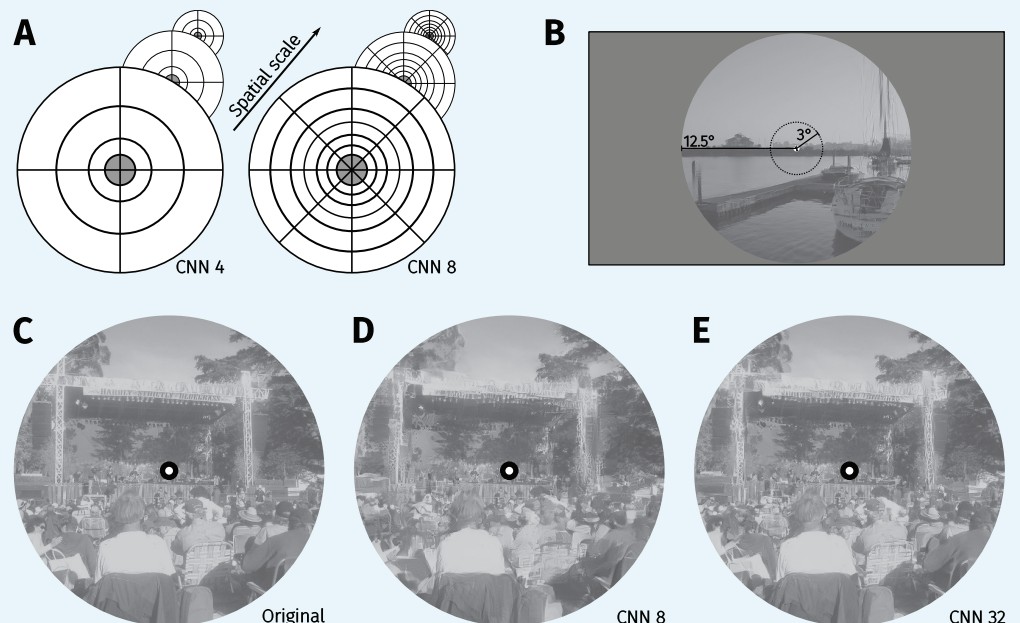

**Appendix 2—figure 1.** Methods for the CNN scene appearance model. (**A**) The average activations in a subset of CNN feature maps were computed over non-overlapping radial and angular pooling regions that increase in area away from the image centre (not to scale), for three spatial scales. Increasing the number of pooling regions (CNN 4 and CNN 8 shown in this example) increases the fidelity of matching to the original image, restricting the range over which distortions can occur. Higher-layer CNN receptive fields overlap the pooling regions, ensuring smooth transitions between regions. The central 3° of the image (grey fill) is fixed to be the original. (**B**) The image radius subtended 12.5°. (**C**) An original image from the MIT1003 dataset. (**D**) Synthesised image matched to the image from C by the CNN 8 pooling model. (**E**) Synthesised image matched to the image from E by the CNN 32 pooling model. Fixating the central bullseye, it should be apparent that the CNN 32 model preserves more information than the CNN 8 model, but that the periphery is nevertheless significantly distorted relative to the original. Images from the MIT 1003 dataset (*Judd et al., 2009*), (https://people.csail.mit.edu/tjudd/WherePeopleLook/index.html) and reproduced under a

In Experiment 1, we tested the discriminability of syntheses generated from the four
pooling models in a set of 400 images that were novel to the participants. Experiment 2
examines the effect of image familiarity by repeatedly presenting a small number of source
images. Experiment 3 tested the effect of cueing spatial attention on performance.

## CNN model methods

### Radial and angular pooling

In the texture synthesis approach of *Gatys et al. (2015)*, spatial information is removed from
the raw CNN activations by computing summary statistics (the Gram matrices of correlations
between feature maps) over the whole image. In the 'foveated' pooling model we present
here, we compute and match the mean of the feature maps (i.e. not the full Gram matrices)
over local image regions by dividing the image into a number of radial and angular pooling
regions (*Appendix 2—figure 1*). The radius defining the border between each radial pooling
region is based on a given number of angular regions $N_\theta$ (which divide the circle evenly) and
given by

$$r_i = r_0 \left( 1 - \frac{\sin(\frac{\pi}{N_\theta}) \cdot 2}{\alpha} \right)^i ,$$

where $r_i$ is the radius of each region $i$, $r_0$ is the outermost radius (set to be half the image size),
and $\alpha$ is the ratio between the radial and angular difference. Radial regions were created for
all $i$ for which $r_i \geq 64$~px, corresponding to the preserved central region of the image (see
below). We set $\alpha = 4$ because at this ratio $N_\theta \approx N_e$ (where $N_e$ is the number of radial regions) for
most $N_\theta$. The value of $N_\theta$ corresponds to the model name used in the paper (e.g. 'CNN 4' uses
$N_\theta = 4$).

We now apply these pooling regions to the activations of the VGG-19 deep CNN
(*Simonyan and Zisserman, 2015*). For a subset of VGG-19 layers (conv1_1, conv2_1, conv3_1,
conv4_1, conv5_1) we compute the mean activation for each feature map $j$ in each layer $l$
within each (radial or angular) pooling region $p$ as

$$w_{pj}^l = \frac{1}{N_l} \sum_{k \in p} (F_{kj}^l),$$

where is $N_l$ the size of the feature map of layer l in pixels and $k$ is the (vectorised) spatial
position in feature map $F_j^l$. The set of all $w_{pj}^l$ provides parameters that specify the foveated
image at a given scale. Note that while the radial and angular pooling region responses are
computed separately, because they are added together to the loss function during
optimisation (see below) they effectively overlap (as depicted in *Appendix 2—figure 1*).

In addition, while the borders of the pooling regions are hard-edged (i.e. pooling regions
are non-overlapping), the receptive fields of CNN units (area of pixels in the image that can
activate a given unit in a feature map) can span multiple pooling regions. This means that the
model parameters of a given pooling region will depend on image structure lying outside the
pooling region (particularly for feature maps in the higher network layers). This encourages
smooth transitions between pooling regions in the synthesised images.

### Multiscale model

In the VGG-19 network, receptive fields of the units are squares of a certain size, and this size
is independent of the input size of the image. That is, given a hypothetical receptive field
centred in the image of size 128 ~ px square, the unit will be sensitive to one quarter of the
image for input size 512 ~ px but half the image for input size 256. Therefore, the same unit in
the network can receive image structure at a different scale by varying the input image size,

and in the synthesis process the low (high) frequency content can be reproduced with greater fidelity by using a small (large) input size.

We leverage this relationship to better capture long-range correlations in image structure (caused by for example edges that extend across large parts of the image) by computing and matching the model statistics over three spatial scales. This is not a controversial idea: for example, the model of *Freeman and Simoncelli (2011)* also computes features in a multiscale framework. How many scales is sufficient?

We evaluated the degree to which the number and combination of scales affected appearance in a psychophysical experiment on authors TW and CF. We matched 100 unique images using seven different models: four single-scale models corresponding to input sizes of 64, 128, 256 and 512 pixels, and three multiscale models in which features were matched at multiple scales ([256, 512 , 128, 256, 512] and [64, 128, 256, 512]). The foveated pooling regions corresponded to the CNN 32 model. Output images were upsampled to the final display resolution as appropriate. We discarded trials for which participants made no response (N = 2) or broke fixation (N = 5), leaving a total of 1393 trials for further analysis.

*Appendix 2—figure 2* shows that participants are sensitive to the difference between model syntheses and original images when features are matched at only a single scale. However, using two or three scales appears to be sufficient to match appearance on average. As a compromise between fidelity and computational tractability, we therefore used three scales for all other experiments on the CNN appearance model. The final model used three networks consisting of the same radial and angular regions described above, computed over sizes 128, 256 and 512 ~ px square. The final model representation $W$ therefore consists of the pooled feature map activations over three scales: $W = \{w^l_{pj,128}, w^l_{pj,256}, w^l_{pj,512}\}$.

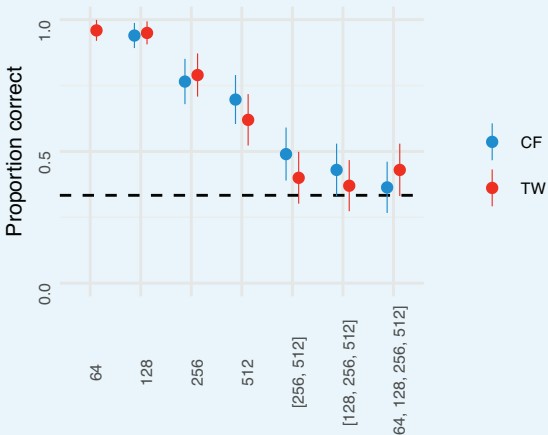

**Appendix 2—figure 2.** Performance for discriminating model syntheses and original scenes for single- and multi-scale models (all with pooling regions corresponding to CNN 32) for participants CF and TW. Points show participant means (error bars show ±2 SEM), dashed line shows chance performance. The multiscale model with three scales produces close-to-chance performance.

DOI: https://doi.org/10.7554/eLife.42512.018

## Gradient descent

As in *Gatys et al. (2015)*, synthesised images are generated using iterative gradient descent, in which the mean-squared distance between the averaged feature maps of the original image and the synthesis is minimized. If $T$ and $W$ are the model representations for the synthesis and the original image respectively, then the loss for each layer is given by

$$L(\vec{x}_t, \vec{x}_g) = \sum_{l,s} \frac{1}{M_{ls} \cdot (N_\theta + N_e)} \sum_{p,j} (W^l_{pjs} - T^l_{pjs})^2,$$

where $\vec{x}_t$ and $\vec{x}_g$ are the vectorised pixels of the original and new image respectively, $M_{l,s}$ is the number of feature maps for layer $l$ in scale $s$. A circular area in the middle of the image (radius $64 \sim$ px) is preserved to be the original image. Tiling pooling regions even for the centre of the image created reasonable syntheses but is prohibitively costly in generation time. To preserve the pixels in the circular area, the initialisation image of the gradient descent is identical to the original image. Outside the central area the gradient descent is initialised with Gaussian noise. The gradient descent used the L-BFGS optimiser (scipy implementation, *Jones et al., 2001*) for 1000 iterations.

## Experiment 1: Discriminability of CNN model syntheses for 400 unique images

This experiment measured whether any of the variants of the CNN scene appearance model could synthesise images that humans could not discriminate from their natural source images, and if so, identify the simplest variant of this model producing metamers. We chose a set of 400 images and had participants discriminate original and model-generated images in a temporal oddity paradigm.

### Methods
The methods for this and the following psychophysical experiments were the same as in the main paper unless otherwise noted.

Participants

Thirteen participants participated in this experiment. Of these, ten participants were recruited via online advertisements and paid 15 Euro for a 1.5 hr testing session; the other three participants were authors AE, TW and CF. One session comprised one experiment using unique images (35 mins) followed by and one of repeated images (see below; 25 mins). All participants signed a consent form prior to participating. Participants reported normal or corrected-to-normal visual acuity. All procedures conformed to Standard 8 of the American Psychological Association's 'Ethical Principles of Psychologists and Code of Conduct' (2010).

Stimuli

We used 400 images (two additional images for authors, see below) from the MIT 1003 database (*Judd et al., 2012*; *Judd et al., 2009*). One of the participants (TW) was familiar with the images in this database due to previous experiments. New images were synthesised using the multiscaled (512 px, 256 px, 128 px) foveated model described above, for four pooling region complexities (4, 8, 16 and 32). An image was synthesised for each of the 400 original images from each model (giving a total stimulus set including originals of 2000).

Procedure

Participants viewed the display from 60 cm; at this distance, pixels subtended approximately 0.024 degrees on average (approximately 41 pixels per degree of visual angle) – note that this is slightly further away than the experiment reported in the primary paper (changed to match the angular subtense used by Freeman and Simoncelli). Images therefore subtended $\approx 12.5°$ at the eye. As in the main paper, the stimuli were presented for 200 ms, with an inter-stimulus interval of 1000 ms, followed by a 1200 ms response window. Feedback was provided by a 100 ms change in fixation cross brightness. Gaze position was recorded during the trial. If the participant moved the eye more than 1.5 degrees away from the fixation spot, feedback signifying a fixation break appeared for $200 \sim$ ms after the response feedback. Prior to the next trial, the state of the participant's eye position was monitored for 50 ms; if the eye position was reported as more than 1.5 degrees away from the fixation spot a recalibration was triggered. The inter-trial interval was 400 ms.

Each unique image was assigned to one of the four models for each participant (counterbalanced). That is, a given image might be paired with a CNN 4 synthesis for one participant and a CNN 8 synthesis for another. Showing each unique image only once ensures that the participants cannot become familiar with the images. For authors, images were divided into only CNN 8, CNN 16 and CNN 32 (making 134 images for each model and 402 trials in total for these participants). To ensure that the task was not too hard for naïve participants we added the easier CNN 4 model (making 100 images for each model version

and 400 trials in total). The experiment was divided into six blocks consisting of 67 trials (65 trials for the last block). After each block a break screen was presented telling the participant their mean performance on the previous trials. During the breaks the participants were free to leave the testing room to take a break and to rest their eyes. At the beginning of each block the eyetracker was recalibrated. Naïve participants were trained to do the task, first using a slower practice of 6 trials and second a correct-speed practice of 30 trials (using five images not part of the stimulus set for the main experiment).

Data analysis

We discarded trials for which participants made no response (N = 81) or broke fixation (N = 440), leaving a total of 4685 trials for further analysis.

Performance at each level of CNN model complexity was quantified using a logistic mixed-effects model. Correct responses were assumed to arise from a fixed effect factor of CNN model (with four levels) plus the random effects of participant and image. The model (in lme4-style notation) was

`correct ~ model + (model | subj) + (model | im_code)`

with `family = Bernoulli('logit')`, and using `contr.sdif` coding for the CNN model factor (**Venables and Ripley, 2002**).

The posterior distribution over model parameters was estimated using weakly-informative priors, which provide scale information about the setting of the model but do not bias effect estimates above or below zero. Specifically, fixed effect coefficients were given Cauchy priors with mean zero and SD 1, random effect standard deviations were given bounded Cauchy priors with mean 0.2 (indicating that we expect some variance between the random effect levels) and SD 1, with a lower-bound of 0 (variances cannot be negative), and correlation matrices were given LKJ(2) priors, reflecting a weak bias against strong correlations (**Stan Development, 2015**). The model posterior was estimated using MCMC implemented in the Stan language (version 2.16.2, **Stan Development Team, 2017**; **Hoffman and Gelman, 2014**), with the model wrapper package brms (version 1.10.2, **Bürkner, 2017**) in the R statistical environment. We computed four chains of 15,000 steps, of which the first 5000 steps were used to tune the sampler; to save disk space we only saved every 5th sample.

## Results and discussion

The CNN 32 model came close to matching appearance on average for a set of 400 images. Discrimination performance for ten naïve participants and three authors is shown in **Appendix 2—figure 3** (lines link individual participant means, based on at least 64 trials, median 94). All participants achieve above-chance performance for the simplest model (CNN 4), indicating that they understood and could perform the task. Performance deteriorates as models match the structure of the original image more precisely.

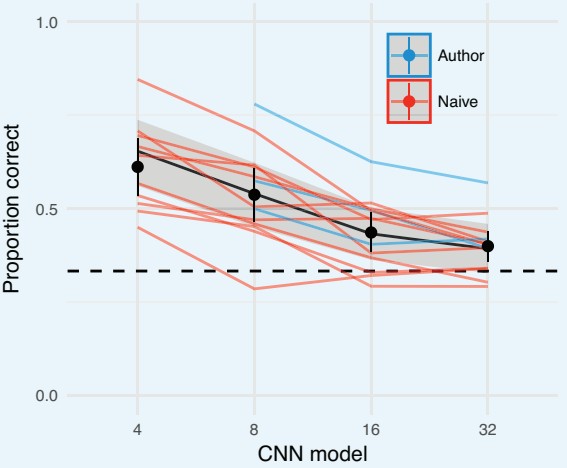

**Appendix 2—figure 3.** The CNN model comes close to matching appearance on average.

Oddity performance as a function of the CNN image model. Points show mean over participants (error bars ±2 SEM), coloured lines link the mean performance of each participant for each pooling model. For most participants, performance falls to approximately chance (dashed horizontal line) for the CNN 32 model. Black line and shaded regions show the mean and 95% credible intervals on the population mean derived from a mixed-effects model.

DOI: https://doi.org/10.7554/eLife.42512.019

To quantify the data, we estimated the posterior distribution of a logistic mixed-effects model with a population-level (fixed-effect) factor of CNN model, whose effect was nested within participants and image (i.e. random effects of participant and image). Regression coefficients coded the difference between successive CNN models, expressed using sequential difference coding from the MASS package (*Venables and Ripley, 2002*), and are presented below as the values of the linear predictor (corresponding to log odds in a logistic model). Mean performance had a greater than 0.99 posterior probability of being lower for CNN 8 than CNN 4 (-0.48, 95% CI $[-0.74, -0.23]$, $p(\beta < 0) > 0.999$), and for CNN 16 being lower than CNN 8 (-0.43, 95% CI $[-0.68, -0.18]$, $p(\beta < 0) = 0.999$); whereas the difference between the 16 and 32 models was somewhat smaller ($-0.17$, 95% CI $[-0.37, 0.03]$, $p(\beta < 0) = 0.951$). Most participants performed close to chance for the CNN 32 model (excluding authors, the population mean estimate had a 0.88 probability of being greater than chance; including authors this value was 0.96). Therefore, the model is capable of synthesising images that are indiscriminable from a large set of arbitrary scenes in our experimental conditions, on average, for naïve participants. However, one participant (author AE) performs noticably better than the others, even for the CNN 32 model. AE had substantial experience with the type of distortions produced by the model but had never seen this set of original images before. Therefore, the images produced by the model are not true metamers, because they do not encapsulate the limits of visible structure for all humans.

## Experiment 2: Image familiarity and learning tested by repeated presentation

It is plausible that familiarity with the images played a role in the results above. That is, the finding that images become difficult on average to discriminate with the CNN 32 model may depend in part on participants having never seen the images before. To investigate the role that familiarity with the source images might play, the same participants as in the experiment above performed a second experiment in which five of the original images from the first experiment were presented 60 times, using 15 unique syntheses per image generated with the CNN 32 model (*Appendix 2—figure 4A*).

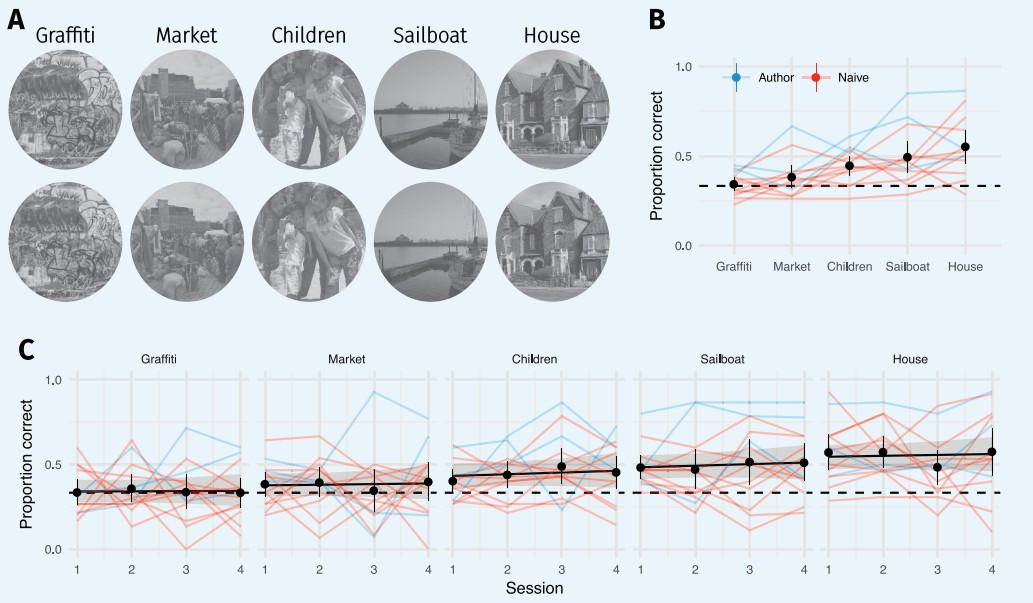

**Appendix 2—figure 4.** Familiarity with original image content did not improve discrimination performance. (**A**) Five original images (top) were repeated 60 times (interleaved over 4 blocks), and observers discriminated them from CNN 32 model syntheses (bottom). (**B**) Proportion of correct responses for each image from A. Some images are easier than others, even for the CNN 32 model. (**C**) Performance as a function of each 75-trial session reveals little evidence that performance improves with repeated exposure. Points show grand mean (error bars show bootstrapped 95% confidence intervals), lines link the mean performance of each observer for each pooling model (based on at least 5 trials; median 14). Black line and shaded region shows the posterior mean and 95% credible intervals of a logistic mixed-effects model predicting the population mean performance for each image. Images from the MIT 1003 dataset (*Judd et al., 2009*, https://people.csail.mit.edu/tjudd/WherePeopleLook/index.html) and reproduced under a CC-BY license (https://creativecommons.org/licenses/by/3.0/) with changes as described in the Materials and methods.
DOI: https://doi.org/10.7554/eLife.42512.020

## Methods

### Participants

The same thirteen participants participated as in Experiment 1.

### Stimuli

We selected five images from the set of 400 and generated 15 new syntheses for each of these images from the CNN 32 model (yielding a stimulus set of 80 images).

### Procedure

Each pairing of unique image (5) and synthesis (15) was shown in one block of 75 trials (pseudo-random order with the restriction that trials from the same source image could never follow one another). Participants performed four such blocks, yielding 300 trials in total (60 repetitions of each original image).

### Data analysis

We discarded trials for which participants made no response (N = 63) or broke fixation (N = 294), leaving a total of 3543 trials for further analysis. Model fitting was as for Experiment 1 above, except that the final posterior was based on four chains of 16,000 steps, of which the first 8000 steps were used to tune the sampler; to save disk space we only saved every 4th sample.

The intercept-only model (assuming only random effects variation but no learning) was specified as

```
correct ~ 1 + (1 | subj) + (1 | im_name)
```

and the learning model was specified as

```
correct ~ session + (session | subj) + (session | im_name)
```

We compare models using an information criterion (LOOIC, *Vehtari et al., 2016*; see also *Gelman et al., 2014*; *McElreath, 2016*) that estimates of out-of-sample prediction error on the deviance scale.

## Results and discussion

While some images (e.g. House) could be discriminated quite well by most participants (*Appendix 2—figure 4B*), others (e.g. Graffiti) were almost indiscriminable from the model image for all participants (posterior probability that the population mean was above chance performance was 0.61 for Graffiti, 0.93 for Market, and greater than 0.99 for all other images). This image dependence shows that even the CNN 32 model is insufficient to produce metamers for arbitrary scenes.

Furthermore, there was little evidence that participants learned over the course of sessions (*Appendix 2—figure 4C*). The population-level linear slope of session number was 0.02, 95% CI $[-0.1, 0.15]$, $p(\beta < 0) = 0.326$, and the LOOIC comparison between the intercept-only model and the model containing a learning term indicated equivocal evidence if random-effects variance was included (LOOIC difference 3.3 in favour of the learning model, SE = 6.1) but strongly favoured the intercept model if only fixed-effects were considered (LOOIC difference $-23.3$ in favour of the intercept model, SE = 1.7). The two images with the most evidence for learning were Children (median slope 0.04, 95% CI $[-0.08, 0.17]$, $p(\beta < 0) = 0.247$) and Sailboat (0.04, 95% CI $[-0.08, 0.17]$, $p(\beta < 0) = 0.269$). Two authors showed some evidence of learning: AE (0.17, 95% CI $[-0.03, 0.37]$, $p(\beta < 0) = 0.047$), and CF (0.22, 95% CI $[0.03, 0.44]$, $p(\beta < 0) = 0.008$). Overall, these results show that repeated image exposures with response feedback did not noticeably improve performance.

## Experiment 3: Spatial cueing of attention

The experiment presented in the primary paper showed that the discriminability of model syntheses depended on the source images, with scene-like images being easier to discriminate from model syntheses than texture-like images for a given image model. This finding was replicated in an ABX paradigm (above) and the general finding of source-image-dependence was corroborated by the data with repeated images (*Appendix 2—figure 3*). One possible reason for this image-dependence could be that participants found it easier to know where to attend in some images than in others, creating an image-dependence not due to the summary statistics per se. Relatedly, *Cohen et al. (2016)* suggest that the limits imposed by an ensemble statistic representation can be mitigated by the deployment of spatial attention to areas of interest. Can the discriminability of images generated by our model be influenced by focused spatial attention?

To probe this possibility we cued participants to a spatial region of the scene before the trial commenced. We computed the mean squared error (MSE) between the original and synthesised images within 12 partially-overlapping wedge-like regions subtending 60°. We computed MSE in both the pixel space (representing the physical difference between the two images) and in the feature space of the fifth convolutional layer (conv5_1) of the VGG-19 network, with the hypothesis that this might represent more perceptually relevant information, and thus be a more informative cue.

We pre-registered the following hypotheses for this experiment (available at http://dx.doi.org/10.17605/OSF.IO/MBGSQ; click on 'View Registration Form'). For the overall effect of cueing (the primary outcome of interest), we hypothesised that

- performance in the Valid:Conv5 condition would be higher than the Uncued condition and
- performance in the Invalid condition would be lower than the Uncued condition

These findings would be consistent with the account that spatial attention can be used to overcome ensemble statistics in the periphery, providing that it is directed to an informative location. This outcome also assumes that our positive cues (Conv5 and Pixels) identify informative locations.

Alternative possibilities are

- if focussed spatial attention cannot influence the 'resolution' of the periphery in this task, then performance in the Valid:Conv5 and Invalid conditions will be equal to the Uncued condition.
- if observers use a global signal ('gist') to perform the task, performance in the Uncued condition would be higher than the Valid:Conv5 and Invalid conditions. That is, directing spatial attention interferes with a gist cue.

Our secondary hypothesis concerns the difference between Valid:Conv5 and Valid:Pixel cues. A previous analysis at the image level (see below) found that conv5 predicted image difficultly slightly better than the pixel space. We therefore predicted that Valid spatial cues based on Conv5 features (Valid:Conv5) should be more effective cues, evoking higher performance, than Valid:Pixel cues.

## Methods

### Participants

We pre-registered (http://dx.doi.org/10.17605/OSF.IO/MBGSQ) the following data collection plan with a stopping rule that depended on the precision (*Kruschke, 2015*). Specifically, we collected data from a minimum of 10 and a maximum of 30 participants, planning to stop in the intermediate range if the 95% credible intervals for the two parameters of interest (population fixed-effect difference between Valid and Uncued, and population fixed-effect difference between Invalid and Uncued) spanned a width of 0.3 or less on the linear predictor scale.

This value was determined as 75% of the width of our 'Region of Practical Equivalence' to zero effect (ROPE), pre-registered as [−0.2, 0.2] on the linear predictor scale (this corresponds to odds ratios of [0.82, 1.22]). We deemed any difference smaller than this value as being too small to be practically important.

As an example, if the performance in one condition is 0.5, then an increase of 0.2 in the linear predictor corresponds to a performance of 0.55. The target for precision was then determined as 75% of the ROPE width, in order to give a reasonable chance for the estimate to lie within the ROPE (*Kruschke, 2015*).

We tested these conditions by fitting the data model (see below) after every participant after the 10th, stopping if the above conditions were met. However, as shown in *Appendix 2—figure 5*, this precision was not met with our maximum of 30 participants, and so we ceased data collection at 30, deeming further data collection beyond our resources for the experiment. Thus our data should be interpreted with the caveat that the desired precision was not reached (though we got close).

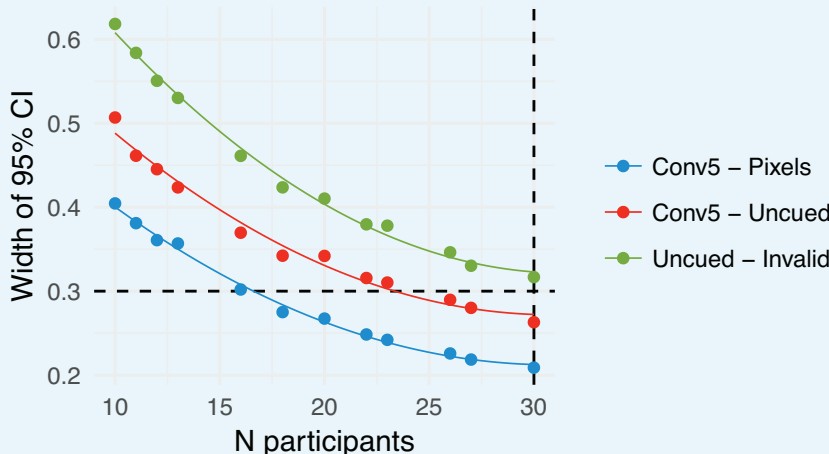

**Appendix 2—figure 5.** Parameter precision as a function of number of participants. (**A**) Width of the 95% credible interval on three model parameters as a function of the number of

participants tested. Points show model fit runs (the model was not re-estimated after every participant due to computation time required). We aimed to achieve a width of 0.3 (dashed horizontal line) on the linear predictor scale, or stop after 30 participants. The Uncued - Invalid parameter failed to reach the desired precision after 30 participants. Lines show fits of a quadratic polynomial as a visual guide.
DOI: https://doi.org/10.7554/eLife.42512.021

An additional five participants were recruited but showed insufficient eyetracking accuracy or training performance (criteria pre-registered). Of the 30, three were lab members unfamiliar with the purpose of the study, the other 27 were recruited online; all were paid 15 Euro for the 1.5 hr testing session. Of these, three participants did not complete the full session due to late arrival, and eyetracking calibration failed in the second last trial block for an additional participant.

Stimuli

This experiment used the same 400 source images and CNN 8 model syntheses as Experiment 1.

Procedure

The procedure for this experiment was as in Experiment 1 with the following exceptions. The same 400 original images were used as in Experiment 1, all with syntheses from the CNN 8 model. A trial began with the presentation of a bright wedge (60 degree angle, Weber contrast 0.25) or circle (radius 2 dva) for 400 ~ ms, indicating a spatial cue (85% of trials) or Uncued trial (15%) respectively (**Appendix 2—figure 6A**). A blank screen with fixation spot was presented for 800 ms before the oddity paradigm proceeded as above. On spatial cue trials, participants were cued to the wedge region containing either the largest pixel MSE between the original and synthesised images (35% of all trials), the largest conv5 MSE (35%), or the *smallest* pixel MSE (an invalid cue, shown on 15% of all trials). Thus, 70% of all trials were valid cues, encouraging participants to make use of the cues rather than learning to ignore them. Participants were also instructed to attend to the cued region on trials where a wedge was shown. For Uncued trials they were instructed to attend globally over the image. Cueing conditions were interleaved and randomly assigned to each unique image for each participant. The experiment was divided into eight blocks of 50 trials. Before the experiment we introduced participants to the task and fixation control with repeated practice sessions of 30 trials (using 30 images not used in the main experiment and with the CNN 4 model syntheses). Participants saw at least 60 and up to 150 practice trials, until they were able to get at least 50% correct and with 20% or fewer trials containing broken fixations or blinks.

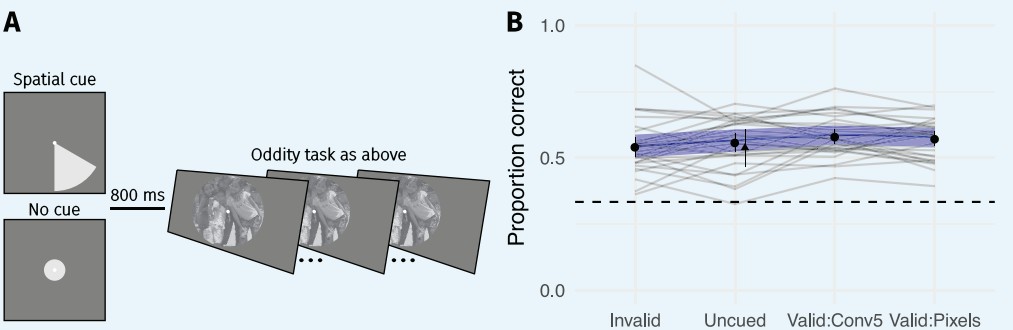

**Appendix 2—figure 6.** Cueing spatial attention has little effect on performance. (**A**) Covert spatial attention was cued to the area of the largest difference between the images (70% of trials; half from conv5 feature MSE; half from pixel MSE) via a wedge stimulus presented before the trial. On 15% of trials the wedge cued an invalid location (smallest pixel MSE), and on 15% of trials no cue was provided (circle stimulus). (**B**) Performance as a function of cueing condition for 30 participants. Points show grand mean (error bars show $\pm 2$ SE), lines link the mean performance of each observer for each pooling model (based on at least 30 trials; median 65). Blue lines and shaded area show the population mean estimate and 95% credible intervals from the mixed-effects model. Triangle in the Uncued condition replots the average

Data analysis

We discarded trials for which participants made no response (N = 141) or broke fixation (N = 1398), leaving a total of 10261 trials for further analysis.

This analysis plan was pre-registered and is available at http://dx.doi.org/10.17605/OSF.IO/MBGSQ (click on 'view registration form'). We seek to estimate three performance differences:

1. The difference between Invalid and Uncued
2. The difference between Valid:Conv5 and Uncued
3. The difference between Valid:Conv5 and Valid:Pixels

The model formula (in lme4-style formula notation) is

```
correct ~ cue + (cue | subj) + (cue | im_code)
```

with `family = Bernoulli('logit')`. The 'cue' factor uses custom contrast coding (design matrix) to test the hypotheses of interest. Specifically, the design matrix for the model above was specified as

|  | $\beta_0$ | $\beta_1$ | $\beta_2$ | $\beta_3$ |
|---|---|---|---|---|
| Invalid | 1 | -1 | 0 | 0 |
| Uncued | 1 | 1 | -1 | 0 |
| Valid:Conv5 | 1 | 0 | 1 | 1 |
| Valid:Pixels | 1 | 0 | 0 | -1 |

Therefore, $\beta_1$ codes Uncued - Invalid, $\beta_2$ codes Valid:Conv5 - Uncued, $\beta_3$ codes Valid:Conv5 - Valid:Pixels and $\beta_0$ codes the Intercept (average performance). Note that the generalised inverse of this matrix was passed to brms (*Venables and Ripley, 2002*).

Each of these population fixed-effects is offset by the random effects of participant (`subj`) and image (`im_code`). We also assume that the offsets for each fixed effect can be correlated (denoted by the single pipe character |). The model thus estimates:

1. Four fixed-effect coefficients. The coefficients coding Valid:Conv5 – Uncued and Uncued – Invalid constitute the key outcome measures of the study. The final coefficient is the analysis of secondary interest.
2. Eight random-effects standard-deviations (four for each fixed-effect, times two for the two random effects).
3. Twelve correlations (six for each pairwise relationship between the fixed-effects, times two for the two random effects).

These parameters were given weakly-informative prior distributions as for Experiment 1 (above): fixed-effects had Cauchy(0, 1) priors, random effect SDs had bounded Cauchy(0.2, 1) priors, and correlation matrices had LKJ(2) priors.

To judge the study outcome we pre-defined a region of practical equivalence (ROPE) around zero effect (0) of [−0.2, 0.2] on the linear predictor scale. This corresponds to odds ratios of [0.82, 1.22]. Our decision rules were then:

- If the 95% credible interval of the parameter value falls outside the ROPE, we consider there to be a credible difference between the conditions.
- If the 95% credible interval of the parameter value falls fully within the ROPE, we consider there to be no practical difference between the conditions. This does not mean that there is no effect, but only that it is unlikely to be large.
- If the 95% credible interval overlaps the ROPE, the data are ambiguous as to the conclusion for our hypothesis. This does not mean that the data give no insight into the direction and

magnitude of any effect, but only that they are ambiguous with respect to our decision criteria.

For more discussion of this approach to hypothesis testing, see (*Kruschke, 2015*).

## Results and discussion

The results of this experiment are shown in *Appendix 2—figure 6B*. While mean performance across conditions was in the expected direction for all effects, no large differences were observed. Specifically, the population-level coefficient estimate on the linear predictor scale for the difference between the Valid:Conv5 cueing condition and the uncued condition was 0.09, 95% CI [−0.05, 0.22], $p(\beta < 0) = 0.1$. Given our decision rules above, the coefficient does not fall wholly within the ROPE and therefore this result is somewhat inconclusive; in general the difference is rather small and so large 'true' effects of spatial cueing are quite unlikely. Similarly, we find no large difference between uncued performance and the invalid cues (0.09, 95% CI [−0.07, 0.25], $p(\beta < 0) = 0.141$). Based on our pre-registered cutoff for a meaningful effect size we conclude that cueing spatial attention in this paradigm results in effectively no performance change.

We further hypothesised that the conv5 cue would be more informative (resulting in a larger performance improvement) than the pixel MSE cue. Note that for 269 of 400 images the conv5 and pixel MSE cued the same or neighbouring wedges, meaning that the power of this experiment to detect differences between these conditions is limited. Consistent with this and contrary to our hypothesis, we find no practical difference between the Valid:Conv5 and Valid:Pixels conditions, 0.04, 95% CI [−0.07, 0.14], $p(\beta < 0) = 0.253$. Note that for this comparison, the 95% credible intervals for the parameter fall entirely within the ROPE, leading us to conclude that there is no practical difference between these conditions in our experiment.

To conclude, our results here suggest that if cueing spatial attention improves the 'resolution' of the periphery, then the effect is very small. *Cohen et al. (2016)* have suggested that an ensemble representation serves to create phenomenal experience of a rich visual world, and that spatial attention can be used to gain more information about the environment beyond simple summary statistics. The results here are contrary to this idea, at least for the specific task and setting we measure here.

Note however that other experimental paradigms may in general be more suitable for assessing the influence of spatial attention than a temporal oddity paradigm. For example, in temporal oddity participants may choose to reallocate spatial attention after the first interval is presented (e.g. on invalid trials pointing at regions of sky). In this respect a single-interval yes-no design (indicating original/synthesis) might be preferable. However, analysis of such data with standard signal detection theory would need to assume that the participants' decision criteria remain constant over trials, whereas it seems likely that decision criteria would depend strongly on the image. To remain consistent with our earlier experiments we nevertheless employed a three-alternative temporal oddity task here; future work could assess whether our finding of minimal influence of spatial cueing depends on this choice.

## Selection of scene- and texture-like images

As discussed in the main paper, we used the results of a pilot experiment (Experiment 3, above) to help select images to provide a strong test of the FS-model. Briefly, 30 observers discriminated 400 images from syntheses produced by the CNN 8 model. Each image was paired with only one unique synthesis (see Experiment 3 above for further details on the experiment).

In an exploratory analysis of that data, we found that there was a large range of difficulty for individual images (as in Experiment 2, above). *Appendix 2—figure 7* shows the image-specific intercepts estimated by the model described above. We examine this rather than the raw data because cueing conditions were randomly assigned to each image for each subject, meaning that the mean performance of the images will depend on this randomisation (though, given our results, the effects are likely to be small). The image-specific intercept from the model estimates the difficulty of each image, statistically marginalising over cueing

condition. While the posterior means for some images were close to chance, and the 95% credible intervals associated with about 100 images overlapped chance performance, approximately 30 images were easily discriminable from their model syntheses, lying above the mean performance for all images with the CNN 8 model.

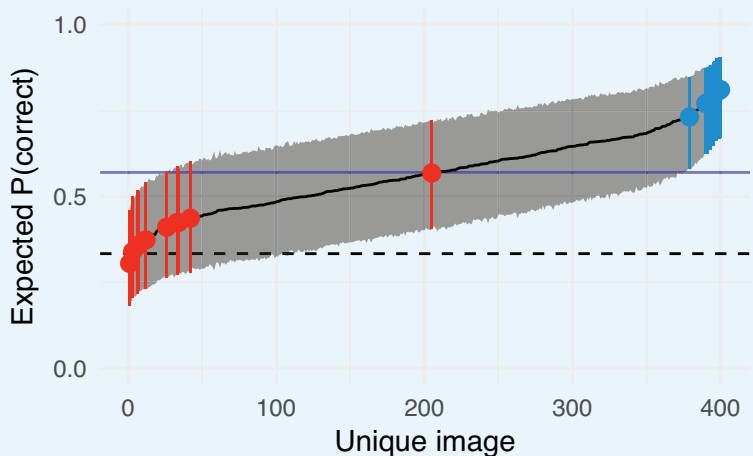

**Appendix 2—figure 7.** Estimated difficulty of each image in Experiment 3 (syntheses with the CNN 8 model) and the images chosen to form the texture- and scene-like categories in the main experiment. Solid black line links model estimates of each image's difficulty (the posterior mean of the image-specific model intercept, plotted on the performance scale). Shaded region shows 95% credible intervals. Dashed horizontal line shows chance performance; solid blue horizontal line shows mean performance. Red and blue points denote the images chosen as texture- and scene-like images in the main experiment respectively. The red point near the middle of the range is the 'graffiti' image from the experiments above.

DOI: https://doi.org/10.7554/eLife.42512.024

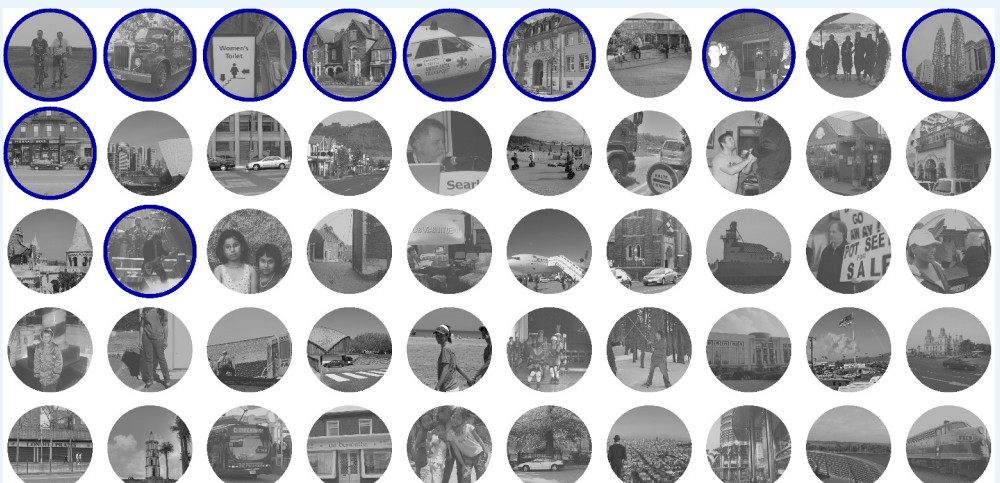

**Appendix 2—figure 8.** The 50 easiest images from *Appendix 2—figure 7* where difficulty increases left-to-right, top-to-bottom. Images chosen for the main experiment as 'scene-like' are circled in blue. Images from the MIT 1003 dataset (*Judd et al., 2009*, https://people.csail.mit.edu/tjudd/WherePeopleLook/index.html) and reproduced under a CC-BY license (https://creativecommons.org/licenses/by/3.0/).

DOI: https://doi.org/10.7554/eLife.42512.025

Our final selection of ten images per category was made by examining the easiest and hardest images from this experiment (*Appendix 2—figure 7*) and selecting ten images we subjectively judged to contain scene-like or texture-like content. The final images used in the first experiment of the main paper are shown in *Appendix 2—figure 7* as coloured points. The 50 easiest and 50 hardest images are shown in *Appendix 2—figures 8,9* respectively.

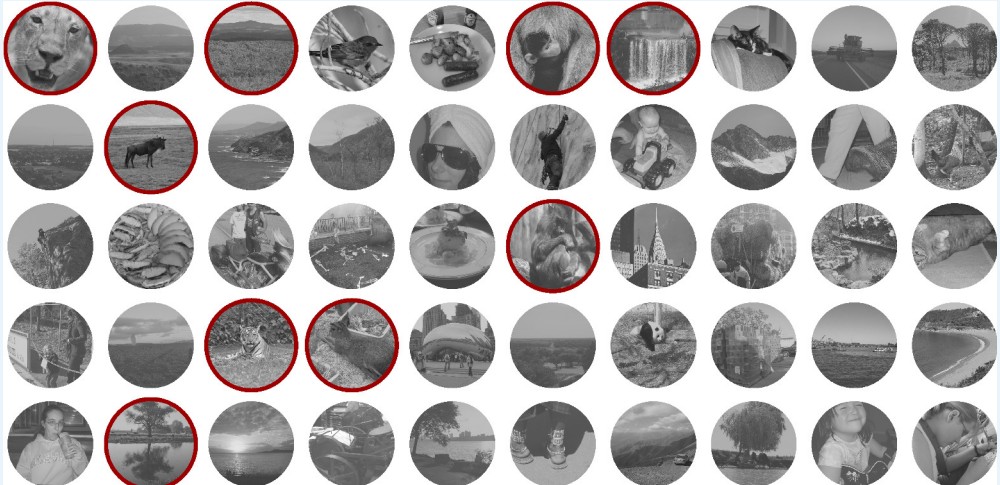

**Appendix 2—figure 9.** The 50 hardest images from *Appendix 2—figure 7* where difficulty decreases left-to-right, top-to-bottom. Images chosen for the main experiment as 'texture-like' are circled in red. Images from the MIT 1003 dataset (*Judd et al., 2009*, https://people.csail. mit.edu/tjudd/WherePeopleLook/index.html) and reproduced under a CC-BY license (https://creativecommons.org/licenses/by/3.0/).
DOI: https://doi.org/10.7554/eLife.42512.026

## Predicting the difficulty of individual images

As shown above, some images are easier than others. We assessed whether an image-based metric considering the difference between original and synthesised images could predict difficulty at the image level. Specifically, we asked whether the mean squared-error (MSE) between the original and synthesised images in two feature spaces (conv5 and pixels) could predict the relative difficulty of the source images. Note that we performed this analysis first on the results of Experiment 1 (*Appendix 2—figure 3*), and that these results were used to inform the hypothesis regarding the usefulness of conv5 vs pixel cues presented in Experiment 3, above. We subsequently performed the same analysis on the data from Experiment 3. We present both analyses concurrently here for ease of reading, but the reader should be aware of the chronological order.

## Methods

We computed the mean squared error between the original and synthesised images in two feature spaces. First, the MSE in the pixel space was used to represent the physical difference at all spatial scales. Second, the difference in feature activations in the conv5 layer of the VGG network was used as an abstracted feature space which may correspond to aspects of human perception (e.g. *Kubilius et al., 2016*, see also *Geirhos et al., 2019*). Both are also correlated with the final value of the loss function from our synthesis procedure. As a baseline we fit a mixed-effects logistic regression containing fixed-effects for the levels of the CNN model and a random effect of observer on all fixed effect terms. As a 'saturated' model (a weak upper bound) we added a random effect for image to the baseline model (that is, each image is uniquely predicted given the available data). Using the scale defined by the baseline and saturated models, we then compared models in which the image-level predictor (pixel or conv5 MSE, standardised to have zero mean and unit variance within each CNN model level) was added as an additional linear covariate to the baseline model. That is, each

image was associated with a scalar value of pixel/conv5 MSE with each synthesis. Additional image-level predictors were compared but are not reported here because they performed similarly or worse than the conv5 or pixel MSE.

As above, we compared the models using the LOOIC information criterion that estimates out-of-sample prediction error on the deviance scale. Qualitatively similar results were found using ten-fold crossvalidation for models fit with penalised maximum-likelihood in lme4.

## Results

For the dataset from Experiment 1, the LOOIC favoured the model containing conv5 MSE over the pixel MSE (LOOIC difference 18.2, SE = 8.3) and the pixel MSE over the baseline model (LOOIC difference 25.3, SE = 10.9)—see *Appendix 2—figure 10A*. The regression weight of the standardised pixel MSE feature fit to all the data was 0.04 (95% credible interval = 0.15–0.07), and the weight of the standardised conv5 feature was 0.04 (0.2–0.11; presented as odds ratios in *Appendix 2—figure 10C*). Therefore, a one standard deviation increase in the conv5 feature produced a slightly larger increase in the linear predictor (and thus the expected probability) than the pixel MSE, in agreement with the model comparison.

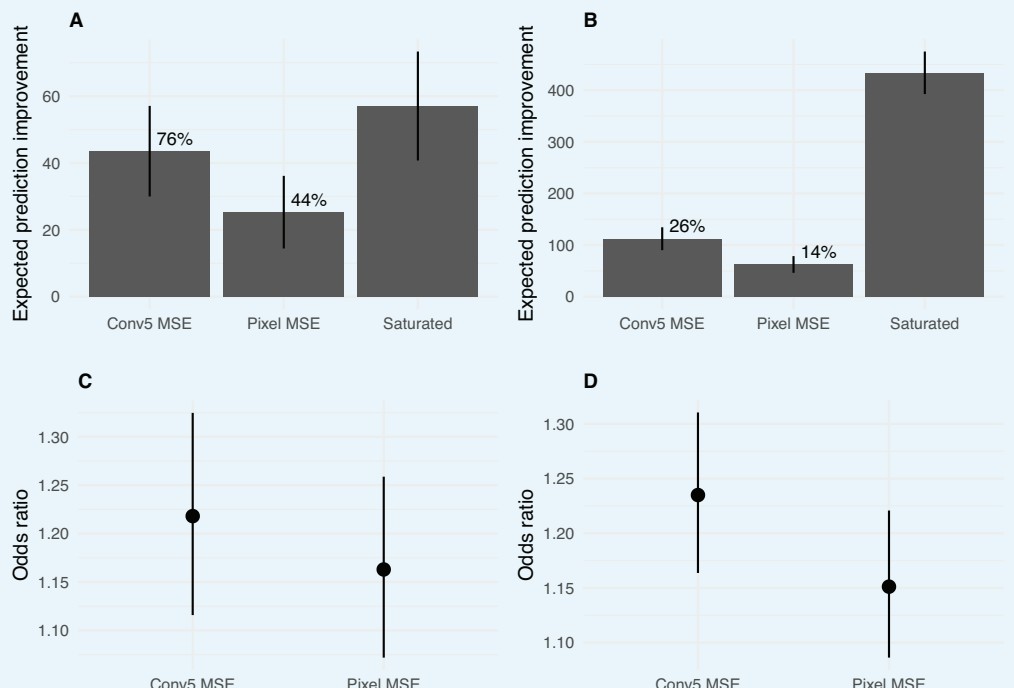

**Appendix 2—figure 10.** Predicting image difficulty using image-based metrics. (**A**) Expected prediction improvement over a baseline model for models fit to the data from Experiment 1 (*Appendix 2—figure 3*), as estimated by the LOOIC (*Vehtari et al., 2016*). Values in deviance units (−2 * log likelihood; higher is better). Error bars show ±2 1 SE. Percentages are expected prediction improvement relative to the saturated model. (**B**) Same as A but for the data from Experiment 3 (*Appendix 2—figure 6*). (**C**) Odds of a success for a one SD increase in the image predictor for data from Experiment 1. Points show mean and 95% credible intervals on odds ratio (exponentiated logistic regression weight). (**D**) As for C for Experiment 3.

DOI: https://doi.org/10.7554/eLife.42512.027

Applying this analysis to the data from Experiment 3 lead to similar results (*Appendix 2—figure 10B,D*). The LOOIC favoured the model containing conv5 MSE over the pixel MSE (LOOIC difference 49.9, SE = 13.3) and the pixel MSE over the baseline model (LOOIC difference 62.4, SE = 16.2). Note that the worse performance of the

image metric models relative to the saturated model (compared to *Appendix 2—figure 10A*) is because the larger data mass in this experiment provides a better constraint for the random effects estimates of image. The regression weight of the standardised pixel MSE feature fit to all the data was 0.03 (95% credible interval = 0.14–0.08), and the weight of the standardised conv5 feature was 0.03 (0.21–0.15).

These results show that the difficulty of a given image can be to some extent predicted from the pixel differences or conv5 differences, suggesting these might prove useful full-reference metrics, at least with respect to the distortions produced by our CNN model.

