## [Decision Letter]

Thank you for submitting your article "Image content is more important than Bouma's Law for scene metamers" for consideration by *eLife*. Your article has been reviewed by three peer reviewers, including Michael Herzog as the Reviewing Editor and Reviewer #1, and the evaluation has been overseen by Timothy Behrens as the Senior Editor. The following individuals involved in review of your submission have agreed to reveal their identity: John Caas (Reviewer #3).

The reviewers have discussed the reviews with one another and the Reviewing Editor has drafted this decision to help you prepare a revised submission.

The authors show that summary statistics models cannot easily explain crowding. In previous studies, only "metameric" images were compared with each other by human observers. In the present study, the original image was compared with two images created by either the FS-model or a deep network. For all scale factors tested, observers could well tell the original from the other images arguing against summary statistic approaches.

The topic is timely, the experiment was conducted in an expert fashion, and the presentation is clear and well structured. However, there are aspects which need to be taken care before a final decision can be made.

Essential revisions:

The failure of FS images to metameric to regular scenes could be a function of many things: The specific size of the pooling regions as a function of eccentricity, the statistics that are measured inside pooling regions, or the contribution of mechanisms for perceptual organization that are not explicitly modeled in the FS approach. The demonstration that metamerism breaks down is important, but the current study does not allow us to clearly distinguish between these various alternatives. Please consider discussing these various accounts of the data in more depth, especially with regard to how we might differentiate between these explanations.

The authors note, "It is the image content, not retinal eccentricity, that is the primary determinant of the visibility of at least some summary statistic distortions." The reviewers agree that global processes may well contribute to the higher spatial resolution observed in this study in response to the real-world images. Exactly what these might be is not specified, nor is the possibility that local interactions may also be at play. Two papers (Robol, Casco and Dakin, 2012; and Van der Burg, Cass and Olivers, 2017) showing that local orthogonal structure is capable of driving perceptual segmentation speak to this. One of the reviewers wonders whether such local interactions might at least partially explain performance differences between the different classes of stimuli (real vs. synthetic; texture-like vs. scene-like)?

Could the authors provide a more detailed description of the criteria they used to categorise the real-world images as either texture-like or scene-like? How and why might metametric sensitivity depend upon this classification criterion?

*Reviewer #1:*

1) The TTM model was used by Freeman and Simoncelli to show that different textures generated from the same image are metamers (i.e., cannot be differentiated by subjects).

2) Wallis et al. correctly claim: if the summary statistics in this model truly mimic the statistics extracted by the brain, subjects should not be able to differentiate texturized images from the original image.

3) However, they show that this is not the case with natural images: subjects can well discriminate between the original and associated textures (using both Simoncelli statistics and statistics from CNN simulations).

4) They claim that the global visual scene must be taken into account and perceptual organization is important. Compressibility by summary statistics depends on image content.

5) They link their results to the debate about the richness of experience. Cohen, Dennett and Kanwisher proposed summary statistics to explain the richness in the periphery. Here, they discuss that summary statistics in fact doesn't seem to be able to do the job.

Although it is not the most groundbreaking experiment ever, I still quite like this paper because:

1) It is good to highlight that texturized images are in fact *not* metamers of the original images.

2) The discussion is interesting bringing together crowding and the debate on using summary statistics to explain the richness of perception in the periphery.

*Reviewer #2:*

The authors describe a single experiment designed to test the hypothesis that natural scenes will be metameric to synthetic scenes created using a model that implements a summary-statistic model of texture-like pooling of visual information in peripheral vision (what the authors call the Freeman-Simoncelli or FS model). Critically, the model allows the scale of pooling regions to increase at different rates as a function of eccentricity, making it possible to examine the specific hypothesis that Bouma's Law (which predicts a scale factor of 0.5) governs pooling across the visual field. The authors rightly point out that the key comparison between natural scenes and synthetic scenes has not been done, and the key contribution here is to do just that. Also, the authors examine performance for scenes that they classify as "texture-like" and "scene-like," which should be equally subject to the metameric treatment if the strong FS-model hypothesis is correct. Briefly, they find that scene type does change the critical scale at which participants can reliably tell natural scenes from synthetic ones, and that the critical scale for scene-like images appears to be a good bit lower than the 0.5 value predicted by Bouma's Law.

The task the authors use is appropriate (a 3AFC oddball judgment applied to images presented in sequence), and I don't have any concerns about the implementation of the FS model. I don't love the fact that the entire stimulus set was comprised of only 20 images (10 per scene type), however. I understand the practical constraints the authors mention regarding the long rendering time for synthetic images, but there's a real concern here: Do these results generalize? The authors emphasize that if we find any image that can be discriminated with a scale factor that's lower than 0.5, then that determines the minimal scale of pooling. If we accept this logic, we have no need to worry about generalizability because all we're looking for is the existence of any image that fits the bill. I think I'd feel better about accepting this logic if the authors' task didn't involve distinguishing two physically identical images from an oddball. While an ideal version of the FS-model would (I think) predict that there should be no measurements at scales coarser than the critical scale that would support discrimination of these images, practically, the model one actually runs is not ideal, and thus the images it produces might not completely meet this standard. To be clear, I'm not saying I disagree with the authors' main conclusions, but I do think it's worth qualifying some of the stronger statements about what you can conclude from 10 images per condition produced by a model that doesn't have clear convergence properties.

More broadly, while I think the discrimination task is a good place to start, I also thought that extending the task beyond the identical-foils version of the design could provide some useful context. For example, suppose instead of 3AFC oddball detection, the authors included a version of the task that was a 2AFC real/synthetic judgment? This obviously is no longer a test for metamerism, but whether or not individuals can reliably tell if a synthesized image looks strange is an important indicator of whether or not they can solve your task without actually trying to discriminate the images (are there two strange images and a non-strange one?). Really what I'd like is a more thorough discussion of how observers might try to accomplish the task when distractors are physically identical, and what that might imply about the strength of the conclusions the authors can draw regarding spatial pooling in the model.

Finally, I like the authors discussion of gestalt properties a great deal, but there's also a competing argument that I think is hard to dismiss: What if the critical scale of 0.5 really is sufficient to make metamers out of natural scenes, but the set of statistics being computed within those regions in the FS-model is inadequate? This is a "God-of-the-gaps" argument to be sure, but again, the issue is that we can't hold a specific implementation of a model to the standard of an ideal. I tend to agree with the authors that there are probably grouping and segmentation processes (or global processes) that contribute to perceptual appearance, but it's an open question whether some of these properties might fall out of a model that incorporated different summary statistics that more closely approximate our own perceptual codes in peripheral vision. Again, this is a place where what I'm looking for is some more nuance – perceptual organization is an interesting thing to think about, but refining our ideas about what texture statistics might be computed in the periphery is also interesting to pursue.

*Reviewer #3:*

This paper uses a 3AFC peripheral distortion detection task to measure the human visual system's spatial resolution for images of real world scenes and synthesised versions of these images. Results indicate that the critical scale with which participants were able to reliably detect distortions was significantly smaller for real world images. A further comparison of the results pertaining to ten "scene-like" and ten "texture-like" images was conducted. Results of this analysis demonstrate that distortions in "scene-like" images ("those containing inhomogeneous structures") could be made at significantly finer scales than are distortions in "texture-like" images ("those containing more homogenous or periodically patterned content").

The authors explain that the critical spatial scale implied by performance regarding the synthesised images is broadly consistent with Freeman and Simoncelli, (2011) V2-like texture pooling model of peripheral visual performance. Given the higher spatial resolution performance implied by performance with the real-world images, and most strikingly, the scene-like images, a convincing argument is mounted that performance in these conditions must be based on either additional information, or at least finer grained information, not available in the FS synthesised images.

These results contribute to the burgeoning literature demonstrating that the deleterious effects of visual clutter can – under certain image-based circumstances – be much smaller than is predicted by Bouma-like pooling. As the authors note, "It is the image content, not retinal eccentricity, that is the primary determinant of the visibility of at least some summary statistic distortions."

This begs the question of 'what types of image content enable the relative high spatial resolution shown here and elsewhere?'. The authors cite some studies relevant to this issue (Saarela et al., 2009; Manassi et al., 2013; Vickery et al., 2009; Herzog et al., 2015), and suggest that "early global segmentation processes influence local perceptual sensitivity" and that "global scene organisation needs to be considered if one wants to capture appearance-yet current models that texturise local regions do not explicitly include perceptual organisation (Herzog et al., 2015)."

Whilst I agree that global processes may well contribute to the higher spatial resolution observed in this study in response to the real-world images, this overlooks the recent finding by Van der Burg, Olivers and Cass (2017) showing that local geometric interactions can profoundly improve peripheral performance (i.e. to break crowding) in densely cluttered heterogenous displays. Specifically, they find that the presence of local T-junction-like orthogonal structure reduces the effects of deleterious pooling to almost zero. I would suggest that these or similar high-order local interactions may be at play here. That result and the possibility that local interactions might play a role here should at least be proposed. Given the (somewhat vaguely defined) criteria used by the authors to categorise the real-world images as either texture-like or scene-like, I wonder what specific information the authors believe the visual system might use extract peripheral real world image content.

---

## [Author Response]

Essential revisions:The failure of FS images to metameric to regular scenes could be a function of many things: The specific size of the pooling regions as a function of eccentricity, the statistics that are measured inside pooling regions, or the contribution of mechanisms for perceptual organization that are not explicitly modeled in the FS approach. The demonstration that metamerism breaks down is important, but the current study does not allow us to clearly distinguish between these various alternatives. Please consider discussing these various accounts of the data in more depth, especially with regard to how we might differentiate between these explanations.

It is correct that our results as presented do not distinguish between possible causes for the failure to match scene appearance of either texture pooling model tested here. We have reworded and expanded our Discussion section to more clearly delineate alternative accounts. Our results do not rule out that some configuration of statistics, pooling or optimisation may exist that succeeds in matching appearance. In fact, we do not see how to falsify this possibility, beyond pointing out that a model using fixed pooling and fixed texture features will not be the most parsimonious description for image content (Discussion section).

To discuss how different explanations might be differentiated, we suggest that future experiments could seek to disrupt or reduce the activity of mechanisms of perceptual organisation using image manipulations such as polarity inversion or image two-toning (subsection “Summary statistics, performance and phenomenology”).

The authors note, "It is the image content, not retinal eccentricity, that is the primary determinant of the visibility of at least some summary statistic distortions." The reviewers agree that global processes may well contribute to the higher spatial resolution observed in this study in response to the real-world images. Exactly what these might be is not specified, nor is the possibility that local interactions may also be at play. Two papers (Robol, Casco and Dakin, 2012; and Van der Burg, Cass and Olivers, 2017) showing that local orthogonal structure is capable of driving perceptual segmentation speak to this. One of the reviewers wonders whether such local interactions might at least partially explain performance differences between the different classes of stimuli (real vs. synthetic; texture-like vs. scene-like)?

We thank the reviewers for pointing us to this relevant literature and highlighting the important distinction between local and global mechanisms that could contribute to our results. We now discuss these papers and the possibility that segmentation and grouping may be critically determined by local interactions in a new subsection “Local vs. global mechanisms”. In addition, we have conducted an analysis of our stimuli in which we used a junction detection algorithm to label different junctions in our images (e.g. L-junctions and T-junctions). We find that junction information is negatively correlated with critical scale, as might be predicted from the local interaction account suggested by reviewer 3 (see Appendix 1—figure 3). However, we leave this analysis as a pilot for future work, which would need to test this in a larger and more controlled image set, as well as testing whether this is specific to junctions per se, or rather (e.g.) to edge density more generally. We thank the reviewers for pointing out this interesting avenue for future work.

Could the authors provide a more detailed description of the criteria they used to categorise the real-world images as either texture-like or scene-like? How and why might metametric sensitivity depend upon this classification criterion?

We have expanded our description of this assignment in both the Results section and Appendix 2—figures 7, 8 and 9. Briefly, we used our subjective judgment of image content combined with the results from a pilot experiment using the CNN 8 model to select images with “scene-like” and “texture-like” content that were easy and hard (respectively) for the CNN 8 model to match. This purposeful selection of images provides a strong test of the FS-model: if the same images were relatively easy and hard for both the FS-model and the CNN-model to match, this would provide evidence that certain image structure is problematic for summary statistic models.

Additional major change:

All reviewers wonder what specifically are the image features that cause the distinction we see between the image classes. We wished to test the role of scene-like and texture-like features more specifically: since these are really properties of local regions of images (and the scale of content therein) rather than natural image photographs as a whole, our classification into these categories at the image level is rather crude.

We therefore included a second experiment (previously reported in the Appendix only) in to the main manuscript (see new Figure 3). We added local, circular texture-like distortions into image regions chosen to be “scene-like” or “texture-like” by author CF, not based on any pilot discrimination results. A control experiment showed that naïve participants showed reasonable agreement with CF’s classification (mean agreement 88.6%; see subsection “Stimuli”). We find that texture-like distortions added to scene-like regions are highly visible, whereas the same types of distortions in texture regions are very difficult to detect (see new Figure 3). This result confirms the importance of “scene-like” vs. “texture-like” structure.

Reviewer #1:1) The TTM model was used by Freeman and Simoncelli to show that different textures generated from the same image are metamers (i.e., cannot be differentiated by subjects).2) Wallis et al. correctly claim: if the summary statistics in this model truly mimic the statistics extracted by the brain, subjects should not be able to differentiate texturized images from the original image.3) However, they show that this is not the case with natural images: subjects can well discriminate between the original and associated textures (using both Simoncelli statistics and statistics from CNN simulations).4) They claim that the global visual scene must be taken into account and perceptual organization is important. Compressibility by summary statistics depends on image content.5) They link their results to the debate about the richness of experience. Cohen, Dennett and Kanwisher proposed summary statistics to explain the richness in the periphery. Here, they discuss that summary statistics in fact doesn't seem to be able to do the job.Although it is not the most groundbreaking experiment ever, I still quite like this paper because:
*1) It is good to highlight that texturized images are in fact* not *metamers of the original images.*
2) The discussion is interesting bringing together crowding and the debate on using summary statistics to explain the richness of perception in the periphery.

We thank the reviewer for their positive comments and their time in assessing the paper.

Reviewer #2:The authors describe a single experiment designed to test the hypothesis that natural scenes will be metameric to synthetic scenes created using a model that implements a summary-statistic model of texture-like pooling of visual information in peripheral vision (what the authors call the Freeman-Simoncelli or FS model). Critically, the model allows the scale of pooling regions to increase at different rates as a function of eccentricity, making it possible to examine the specific hypothesis that Bouma's Law (which predicts a scale factor of 0.5) governs pooling across the visual field. The authors rightly point out that the key comparison between natural scenes and synthetic scenes has not been done, and the key contribution here is to do just that. Also, the authors examine performance for scenes that they classify as "texture-like" and "scene-like," which should be equally subject to the metameric treatment if the strong FS-model hypothesis is correct. Briefly, they find that scene type does change the critical scale at which participants can reliably tell natural scenes from synthetic ones, and that the critical scale for scene-like images appears to be a good bit lower than the 0.5 value predicted by Bouma's Law.The task the authors use is appropriate (a 3AFC oddball judgment applied to images presented in sequence), and I don't have any concerns about the implementation of the FS model. I don't love the fact that the entire stimulus set was comprised of only 20 images (10 per scene type), however. I understand the practical constraints the authors mention regarding the long rendering time for synthetic images, but there's a real concern here: Do these results generalize?

We agree that it would be nice to test generalisability, but as the reviewer notes, the computation time to generate a number of scale factors for a large number of images is infeasible. However, as we mention in the paper (Discussion section), a recent preprint provides some evidence that at least part of this result is not limited to only our 20 images. Deza et al. (2017) reported percent correct performance in an ABX task for FS syntheses generated at scale factors of 0.5 for 50 original images. For all six of their naive observers, performance in discriminating original and synthesised images lay above chance (50%), and for four of six observers performance lay above 70%. This result corroborates our finding that FS syntheses can be discriminated from original images at scale factors of 0.5, but does not allow the estimation of critical scales as we do here.

The authors emphasize that if we find any image that can be discriminated with a scale factor that's lower than 0.5, then that determines the minimal scale of pooling. If we accept this logic, we have no need to worry about generalizability because all we're looking for is the existence of any image that fits the bill. I think I'd feel better about accepting this logic if the authors' task didn't involve distinguishing two physically identical images from an oddball. While an ideal version of the FS-model would (I think) predict that there should be no measurements at scales coarser than the critical scale that would support discrimination of these images, practically, the model one actually runs is not ideal, and thus the images it produces might not completely meet this standard.

The reviewer is correct that there is a potential discrepancy between idealised predictions and models in practice. The idealised FS logic would predict that if a hypothetical image had a critical scale of 0.3, then performance should remain at chance for any lower scale factors (e.g. 0.15), but the images should be discriminable for larger (coarser) scales. It is possible that there are implementational details in the models that might cause artifacts that violate these assumptions (for example, if the image synthesis produces images with stronger artifacts at low scale factors). We have run control experiments (Appendix 1—figure 1 and Appendix 1—figure 2) ruling out two possible artifacts as being sole explanations for our result. We agree with the general point that an idealised model can never be tested. Our experiments cannot rule out the possibility that a better implementation of the model may succeed in creating metamers, but we speculate that more will be required. We now try to be clearer on this point in the manuscript (see below).

To be clear, I'm not saying I disagree with the authors' main conclusions, but I do think it's worth qualifying some of the stronger statements about what you can conclude from 10 images per condition produced by a model that doesn't have clear convergence properties.

In response to this comment and the reviewer’s call for more nuance (below), we have been more explicit that our result cannot rule out the existence of some pooling model that could match appearance for all images at V2 scaling (Discussion section).

More broadly, while I think the discrimination task is a good place to start, I also thought that extending the task beyond the identical-foils version of the design could provide some useful context. For example, suppose instead of 3AFC oddball detection, the authors included a version of the task that was a 2AFC real/synthetic judgment? This obviously is no longer a test for metamerism, but whether or not individuals can reliably tell if a synthesized image looks strange is an important indicator of whether or not they can solve your task without actually trying to discriminate the images (are there two strange images and a non-strange one?). Really what I'd like is a more thorough discussion of how observers might try to accomplish the task when distractors are physically identical, and what that might imply about the strength of the conclusions the authors can draw regarding spatial pooling in the model.

We are not sure that we understand the reviewer’s point here and would be happy to re-address the concern if clarification is required. To summarise the point as we understand it: the reviewer suggests it is plausible that synthesised images would be “acceptable” as real in a single-interval task (i.e. with no direct discrimination). That is, in a task where one image is shown per trial (either an original or a synthesised image) and observers label that image as “real” or “synthesised”, the reviewer suggests that observers may have low sensitivity to the true class label but have a bias to report “real”.

This is an interesting question which speaks to acceptable distortion quality. Potentially it is the case that synthesised images that can be told apart from real images in a discrimination task (e.g. our oddity design) might appear “non-strange” when no other reference is given. Such a result would suggest that a certain level of pooling discards an “acceptable” amount of information. However, what is an “acceptable” level of distortion is a separate question from the one we are concerned with here: what information is irretrievably discarded by the visual system. A discrimination experiment is a more direct test of the latter.

Finally, I like the authors discussion of gestalt properties a great deal, but there's also a competing argument that I think is hard to dismiss: What if the critical scale of 0.5 really is sufficient to make metamers out of natural scenes, but the set of statistics being computed within those regions in the FS-model is inadequate? This is a "God-of-the-gaps" argument to be sure, but again, the issue is that we can't hold a specific implementation of a model to the standard of an ideal.

We agree that our results cannot exclude the possibility that a summary statistic model exists that could yield scalings of 0.5. We now make this clearer in our revised Discussion section. However, that three of the known and image-computable summary statistic models (FS, our own CNN model, and the NeuroFovea model of Deza et al., 2017) fail to match appearance supports our claim that summary statistic models with fixed pooling regions may be inadequate to match appearance.

Thus, we cannot rule out that such a model exists, only that none of the known ones work. We clearly label our discussion of Gestalt properties “speculation” (Discussion section) for this reason.

I tend to agree with the authors that there are probably grouping and segmentation processes (or global processes) that contribute to perceptual appearance, but it's an open question whether some of these properties might fall out of a model that incorporated different summary statistics that more closely approximate our own perceptual codes in peripheral vision. Again, this is a place where what I'm looking for is some more nuance – perceptual organization is an interesting thing to think about, but refining our ideas about what texture statistics might be computed in the periphery is also interesting to pursue.

See our response to the editor’s summary letter, above. We have reworded our Discussion section to make it clearer that we cannot rule out some unknown texture statistics as an explanation, and we have expanded our discussion of local and global mechanisms. However, we also re-iterate our general speculation that no fixed texture representation computed over fixed pooling regions can parsimoniously account for image appearance, for it will have to preserve information that could be discarded (Discussion section).

Reviewer #3:This paper uses a 3AFC peripheral distortion detection task to measure the human visual system's spatial resolution for images of real world scenes and synthesised versions of these images. Results indicate that the critical scale with which participants were able to reliably detect distortions was significantly smaller for real world images. A further comparison of the results pertaining to ten "scene-like" and ten "texture-like" images was conducted. Results of this analysis demonstrate that distortions in "scene-like" images ("those containing inhomogeneous structures") could be made at significantly finer scales than are distortions in "texture-like" images ("those containing more homogenous or periodically patterned content").The authors explain that the critical spatial scale implied by performance regarding the synthesised images is broadly consistent with Freeman and Simoncelli, (2011) V2-like texture pooling model of peripheral visual performance. Given the higher spatial resolution performance implied by performance with the real-world images, and most strikingly, the scene-like images, a convincing argument is mounted that performance in these conditions must be based on either additional information, or at least finer grained information, not available in the FS synthesised images.These results contribute to the burgeoning literature demonstrating that the deleterious effects of visual clutter can – under certain image-based circumstances – be much smaller than is predicted by Bouma-like pooling. As the authors note, "It is the image content, not retinal eccentricity, that is the primary determinant of the visibility of at least some summary statistic distortions."This begs the question of 'what types of image content enable the relative high spatial resolution shown here and elsewhere?'. The authors cite some studies relevant to this issue (Saarela et al., 2009; Manassi et al., 2013; Vickery et al., 2009; Herzog et al., 2015), and suggest that "early global segmentation processes influence local perceptual sensitivity" and that "global scene organisation needs to be considered if one wants to capture appearance-yet current models that texturise local regions do not explicitly include perceptual organisation (Herzog et al., 2015)."Whilst I agree that global processes may well contribute to the higher spatial resolution observed in this study in response to the real-world images, this overlooks the recent finding by Van der Burg, Olivers and Cass (2017) showing that local geometric interactions can profoundly improve peripheral performance (i.e. to break crowding) in densely cluttered heterogenous displays. Specifically, they find that the presence of local T-junction-like orthogonal structure reduces the effects of deleterious pooling to almost zero. I would suggest that these or similar high-order local interactions may be at play here. That result and the possibility that local interactions might play a role here should at least be proposed. Given the (somewhat vaguely defined) criteria used by the authors to categorise the real-world images as either texture-like or scene-like, I wonder what specific information the authors believe the visual system might use extract peripheral real world image content.

We thank the reviewer for these insightful comments and for pointing us to this relevant literature. Please see our comments to the main revisions: we have now included a greatly expanded discussion of these issues, as well as a pilot analysis of junction information in our images. This analysis shows that, consistent with the reviewer’s suggestion, images with lower critical scales tended to contain more junction information. This result will be interesting to follow up more systematically in future work.